# Pullback Flow Matching on Data Manifolds

## Abstract

We propose Pullback Flow Matching (PFM), a novel framework for generative modeling on data manifolds. Unlike existing methods that assume or learn restrictive closed-form manifold mappings for training Riemannian Flow Matching (RFM) models, PFM leverages pullback geometry and isometric learning to preserve the underlying manifold's geometry while enabling efficient generation and precise interpolation in latent space. This approach not only facilitates closed-form mappings on the data manifold but also allows for designable latent spaces, using assumed metrics on both data and latent manifolds. By enhancing isometric learning through Neural ODEs and proposing a scalable training objective, we achieve a latent space more suitable for interpolation, leading to improved manifold learning and generative performance. We demonstrate PFM's effectiveness through applications in synthetic data, protein dynamics and protein sequence data, generating novel proteins with specific properties. This method shows strong potential for drug discovery and materials science, where generating novel samples with specific properties is of great interest.

## 1 Introduction

Since the rise of machine learning in the scientific domain, researchers have focused on developing larger models trained on increasingly massive datasets, as in weather forecasting (Bodnar et al., 2024) and protein structure prediction (Hayes et al., 2024). However, relying on such scaling laws is not feasible in many scientific fields where data is limited and precise modeling of physical phenomena is crucial. In such cases, incorporating prior knowledge about the geometry of the data as an inductive bias enables models to make accurate interpolations between data points, which is essential for reliable predictions and realistic representations of complex systems. Current methods, however, lack the mathematical foundations to accurately interpolate in latent space

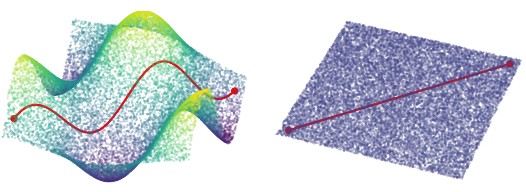

*Data manifold* ($\mathcal{D}$)  *Latent manifold* ($\mathcal{M}$)

Figure 1: An example of isometric learning, where the goal is to create a latent space that allows for interpolation. The shortest paths (in red) on the data manifold $\mathcal{D} \subset \mathbb{R}^3$ correspond to the shortest paths on the latent manifold $\mathcal{M}$.

(Arvanitidis et al., 2017) and do not capture the underlying geometric structure of the data (Wessels et al., 2024). Our goal is to develop mappings that enable precise interpolation in latent space, leveraging geometry as an inductive bias to facilitate efficient and accurate generation on data manifolds, thereby advancing the ability to model complex physical phenomena with limited data.

We consider modeling the data under the manifold hypothesis, which states that high-dimensional data lies on a lower dimensional manifold. This has been successfully applied to several downstream tasks in various fields across the scientific domain (Vanderplas & Connolly, 2009; Dsilva et al., 2016; Noé & Clementi, 2017). Modeling the data in its intrinsic dimension allows for efficient analysis Diepeveen et al. (2024) and generation Rombach et al. (2022). Furthermore, accurately capturing the geometry of the data manifold in the learning problem has shown to improve several down-stream tasks such as clustering (Ghojogh et al., 2022), classification (Kaya & Bilge, 2019; Hauberg et al., 2012) and generation (Arvanitidis et al., 2020; Sun et al., 2024).

One way to achieve a latent manifold that supports interpolation is to have a structured Riemannian geometry, such as the one from *pullback geometry*, which provides closed-form manifold mappings (Diepeveen et al., 2024). This requires constructing an invertible and differentiable mapping—*diffeomorphism*—between the data manifold and the latent manifold. Interpolation on manifolds is then performed through geodesics, shortest paths, and thus to achieve our goal we require geodesics on the data manifold to match geodesics on the latent manifold. This motivates our consideration of *isometries*, that is, metric-preserving diffeomorphisms $\varphi$. These mappings preserve the distances of points on the data manifold on the latent manifold, and thereby ensure proper interpolation.

**Related-Work**. In the literature, low-dimensional generation and generation on manifolds have typically been addressed as separate problems. Low-dimensional approaches, such as latent diffusion (Rombach et al., 2022) or latent flow-matching (Dao et al., 2023), often overlook the geometric structure of the data, leading to inaccuracies in tasks requiring a faithful representation of the underlying manifold. Conversely, manifold generation methods either assume geodesics on the data manifold for simulation-free training (Chen & Lipman, 2024)—an approach flawed when closed-form mappings are unavailable—or attempt to learn a metric that forces the generative trajectories to have data support (Kapusniak et al., 2024).

Using a pullback framework presents challenges, such as task-specific learning problems that limit generality and prevent the learning of isometries across broader data manifolds Cuzzolin (2008); Gruffaz et al. (2021); Lebanon (2006). Geometrically regularized latent space methods, like Lee et al. (2022) and Duque et al. (2022), work in practice but lack solid mathematical grounding in isometries, particularly guaranteeing diffeomorphism in architectural design. Diepeveen (2024) addresses isometry challenges with a more general mathematically grounded framework, but its learning objective's expressivity and computational feasibility limit its application to high-dimensional real-world datasets.

Our approach bridges these gaps by modeling data on a lower-dimensional latent manifold with known geometry through a diffeomorphisms parameterized and trained in a scalable and expressive way. By doing so we preserve the intrinsic properties of the data manifold and enable accurate and efficient generation through simulation-free training.

**Contributions.** We propose Pullback Flow Matching (PFM), a novel framework for latent manifold learning and generation through isometries. This method respects the geometry of the data manifold, even when closed-form manifold mappings are not available. Second, learning can be performed in the intrinsic dimension of the data manifold resulting in efficient and effective learning of the generative model with fewer parameters. Building on Diepeveen (2024), we leverage pullback geometry to define a new metric on the entire ambient space, $\mathbb{R}^d$, by learning an isometry $\varphi$ that preserves the geometric structure of the data manifold $\mathcal{D}$ on the latent manifold $\mathcal{M}$. We use the corresponding metric of the assumed latent manifold $\mathcal{M}$ to perform Riemannian Flow Matching (RFM) on the latent manifold that supports interpolation. Our contributions are as follows:

1. We introduce PFM, a novel framework that enables accurate and efficient data generation on manifolds. PFM leverages the pullback geometry to preserve the underlying geometric structure of the data manifold within the latent space, facilitating precise interpolation and generation.

2. We improve the parameterization of diffeomorphisms, used to learn isometries, in both expressiveness and training efficiency through neural ordinary differential equations (Neural ODEs).

3. We introduce a scalable and stable isometric learning objective. This objective relies solely on a distance measure on the data manifold, simplifying the training process compared to Diepeveen (2024) while maintaining geometric fidelity.

4. We demonstrate our methods' effectiveness through experiments on synthetic data, high-dimensional molecular dynamics data, and experimental peptide sequences. Our framework utilizes *designable latent spaces* to generate novel proteins with specific properties closely matching reference samples. This directed generation showcases the significant applicability of isometric learning and PFM in accurate physical modeling and interpolation, advancing generative modeling techniques in drug discovery and materials science. [1]

---

[1]The anonymized code for the experiments on the synthetic data is available in the supplementary material.

## 2   NOTATION

We give a brief summary of the notation used in the paper, and give a more extensive background on Riemannian and pullback geometry in Appendix A.

A *manifold* $\mathcal{M}$ is a topological space that locally resembles Euclidean space. A $d$-dimensional manifold $\mathcal{M}$ around a point $\boldsymbol{p} \in \mathcal{M}$ is described by a *chart* $\psi : U \to \mathbb{R}^d$, where $U \subseteq \mathcal{M}$ is a neighborhood of $\boldsymbol{p}$. The chart provides a local coordinate system for the manifold. The *tangent space* at a point $\boldsymbol{p} \in \mathcal{M}$, denoted $\mathcal{T}_{\boldsymbol{p}}\mathcal{M}$, is the vector space of all tangent vectors at that point.

A smooth manifold $\mathcal{M}$ equipped with a *Riemannian metric* is called a *Riemannian manifold* and is denoted by $(\mathcal{M}, (\cdot, \cdot)^{\mathcal{M}})$. The Riemannian metric $(\cdot, \cdot)^{\mathcal{M}}$ is a smoothly varying inner product defined on the tangent spaces $\mathcal{T}_{\boldsymbol{p}}\mathcal{M}$ for all points $\boldsymbol{p} \in \mathcal{M}$, and it defines lengths and angles on the manifold. A *geodesic*, $\gamma_{\boldsymbol{p},\boldsymbol{q}}(t)$ is the shortest path between two points $\boldsymbol{p}, \boldsymbol{q} \in \mathcal{M}$, generalizing the notion of a straight line in Euclidean space.

The *exponential map* $\exp_{\boldsymbol{p}} : \mathcal{T}_{\boldsymbol{p}}\mathcal{M} \to \mathcal{M}$ maps a tangent vector $\Xi_{\boldsymbol{p}}$ to a point on the manifold by following the geodesic in the direction of $\Xi_{\boldsymbol{p}}$ starting from $\boldsymbol{p}$. The inverse of the exponential map is the *logarithmic map*, denoted by $\log_{\boldsymbol{p}} : \mathcal{M} \to \mathcal{T}_{\boldsymbol{p}}\mathcal{M}$, which returns the tangent vector corresponding to a given point on the manifold.

In this work, we consider a $d$-dimensional Riemannian manifold $(\mathcal{M}, (\cdot, \cdot)^{\mathcal{M}})$, and a smooth diffeomorphism $\varphi : \mathbb{R}^d \to \mathcal{M}$, such that $\varphi(\mathbb{R}^d) \subseteq \mathcal{M}$ is geodesically convex, meaning that any pair of points within this subset are connected by a unique geodesic. This mapping allows us to pullback the geometric structure of $\mathcal{M}$ to $\mathbb{R}^d$ by defining the *pullback metric* on $\mathbb{R}^d$. Specifically, for tangent vectors $\Xi_{\boldsymbol{p}}, \Phi_{\boldsymbol{p}} \in \mathcal{T}_{\boldsymbol{p}}\mathbb{R}^d$, the pullback metric is defined as

$$(\Xi_{\boldsymbol{p}}, \Phi_{\boldsymbol{p}})^{\varphi} := (\varphi_*[\Xi_{\boldsymbol{p}}], \varphi_*[\Phi_{\boldsymbol{p}}])^{\mathcal{M}}_{\varphi(\boldsymbol{p})}, \tag{1}$$

where $\varphi_*$ is the pushforward of tangent vectors under $\varphi$. Through this construction, various geometric objects in $\mathcal{M}$, such as distances and geodesics, can be expressed in terms of their counterparts in $\mathbb{R}^d$ with respect to the pullback metric. The distance function $d^{\varphi}_{\mathbb{R}^d} : \mathbb{R}^d \times \mathbb{R}^d \to \mathbb{R}$ on $\mathbb{R}^d$ with the pullback metric is given by,

$$d^{\varphi}_{\mathbb{R}^d}(\boldsymbol{x}_i, \boldsymbol{x}_j) = d_{\mathcal{M}}(\varphi(\boldsymbol{x}_i), \varphi(\boldsymbol{x}_j)), \tag{2}$$

where $d_{\mathcal{M}}$ denotes the Riemannian distance on $\mathcal{M}$. The length-minimizing geodesic connecting $\boldsymbol{x}_i$ and $\boldsymbol{x}_j$ in $\mathbb{R}^d$ with respect to the pullback metric $\gamma^{\varphi}_{\boldsymbol{x}_i, \boldsymbol{x}_j} : [0, 1] \to \mathbb{R}^d$ is given by,

$$\gamma^{\varphi}_{\boldsymbol{x}_i, \boldsymbol{x}_j}(t) = \varphi^{-1}(\gamma^{\mathcal{M}}_{\varphi(\boldsymbol{x}_i), \varphi(\boldsymbol{x}_j)}(t)), \tag{3}$$

here $\gamma^{\mathcal{M}}$ denotes the geodesic in $\mathcal{M}$ connecting $\varphi(\boldsymbol{x}_i)$ and $\varphi(\boldsymbol{x}_j)$. This enables computation of geodesics and distances in $\mathbb{R}^d$ using the geometry of $\mathcal{M}$, as stated in Prop. 2.1 of Diepeveen (2024).

In this paper we will assume the standard Euclidean metric $(\cdot, \cdot)_2$ and a Euclidean latent manifold $\mathcal{M} = \mathbb{R}^d$. Hence, the pullback metric will be defined as

$$(\Xi_{\boldsymbol{p}}, \Phi_{\boldsymbol{p}})^{\varphi} := (\varphi_*[\Xi_{\boldsymbol{p}}], \varphi_*[\Phi_{\boldsymbol{p}}])^{\mathbb{R}^d}_{\varphi(\boldsymbol{p})}. \tag{4}$$

We will calculate distances on the latent manifold $\mathcal{M} = \mathbb{R}^d$ through,

$$d^{\varphi}_{\mathbb{R}^d}(\boldsymbol{x}_i, \boldsymbol{x}_j) = \|\varphi(\boldsymbol{x}_i) - \varphi(\boldsymbol{x}_j)\|_2, \tag{5}$$

and the geodesic calculation will boil down to

$$\gamma^{\varphi}_{\boldsymbol{x}_i, \boldsymbol{x}_j}(t) = \varphi^{-1}(\varphi(\boldsymbol{x}_i)(1-t) + t\varphi(\boldsymbol{x}_j)). \tag{6}$$

An example of a pullback geodesic $\gamma^{\varphi}_{\boldsymbol{x}_i, \boldsymbol{x}_j}(t)$ on the data manifold based on a geodesic on a latent Euclidean manifold can be viewed in Figure 2.

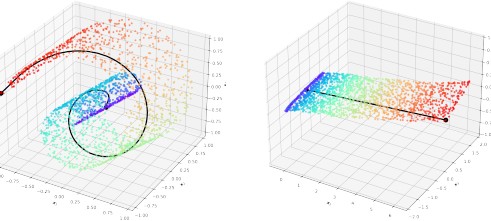

*Data manifold* ($\mathcal{D}$)      *Latent manifold* ($\mathcal{M}$)

Figure 2: Isometric learning for the rotated Swiss roll in 3D. The learned geodesic path (in black) on the data manifold $\mathcal{D} \subset \mathbb{R}^3$ correspond to the shortest paths on the latent manifold $\mathcal{M} = \mathbb{R}^3$.

## 3 PULLBACK FLOW MATCHING

We propose *Pullback Flow Matching (PFM)*, a novel framework for generative modeling on data manifolds using *pullback geometry*. Our goal is to transform samples from a simple distribution $x_0 \sim p$ on the data manifold $\mathcal{D}$ into a complex target distribution $x_1 \sim q$, also on $\mathcal{D}$. Ideally, we would perform this transformation using *Riemannian Flow Matching (RFM)*, see Appendix A for a summary, on $(\mathcal{D}, (\cdot, \cdot)^{\mathcal{D}})$ by optimizing the objective from Chen & Lipman (2024),

$$\mathcal{L}_{RFM}(\boldsymbol{\eta}) = \mathbb{E}_{t, q(\boldsymbol{x}_1), p(\boldsymbol{x}_0)} \left( \left\| v_t \left( \gamma_{\boldsymbol{x}_1, \boldsymbol{x}_0}^{\mathcal{D}}(t); \boldsymbol{\eta} \right) - \dot{\gamma}_{\boldsymbol{x}_1, \boldsymbol{x}_0}^{\mathcal{D}}(t) \right\|_{\gamma_{\boldsymbol{x}_1, \boldsymbol{x}_0}^{\mathcal{D}}(t)}^{\mathcal{D}} \right)^2, \tag{7}$$

where $\boldsymbol{\eta}$ represents the learnable parameters of the parameterized vector field $v_t(\boldsymbol{x}; \boldsymbol{\eta})$. Solving this objective on data manifolds becomes intractable as the training of RFM is no longer simulation-free (Chen & Lipman, 2024). Existing methods address this challenge by employing restrictive and computationally intensive manifold mappings (Kapusniak et al., 2024). We overcome this limitation by defining a new metric on the ambient space $\mathbb{R}^d$ using the *pullback metric* (Diepeveen, 2024) and assume a learned *isometry* $\varphi_{\boldsymbol{\theta}}$ that approximates geodesics $\gamma^{\varphi_{\boldsymbol{\theta}}}$ on $(\mathbb{R}^d, (\cdot, \cdot)^{\varphi_{\boldsymbol{\theta}}})$ to those $\gamma^{\mathcal{D}}$ on $(\mathcal{D}, (\cdot, \cdot)^{\mathcal{D}})$. Rewriting the RFM objective under the pullback framework yields the objective,

$$\mathcal{L}_{PFM}(\boldsymbol{\eta}) = \mathbb{E}_{t, q(\boldsymbol{x}_1), p(\boldsymbol{x}_0)} \left( \left\| v_t \left( \gamma_{\boldsymbol{x}_1, \boldsymbol{x}_0}^{\varphi_{\boldsymbol{\theta}}}(t); \boldsymbol{\eta} \right) - \dot{\gamma}_{\boldsymbol{x}_1, \boldsymbol{x}_0}^{\varphi_{\boldsymbol{\theta}}}(t) \right\|_{\gamma_{\boldsymbol{x}_1, \boldsymbol{x}_0}^{\varphi_{\boldsymbol{\theta}}}(t)}^{\varphi_{\boldsymbol{\theta}}} \right)^2, \tag{8}$$

By applying Equation 3, we reformulate the PFM objective in terms of manifold mappings on $\mathcal{M}$,

$$\mathcal{L}_{PFM}(\boldsymbol{\eta}) =$$
$$\mathbb{E}_{t, q(\boldsymbol{x}_1), p(\boldsymbol{x}_0)} \left( \left\| v_t \left( \gamma_{\varphi_{\boldsymbol{\theta}}(\boldsymbol{x}_1), \varphi_{\boldsymbol{\theta}}(\boldsymbol{x}_0)}^{\mathcal{M}}(t); \boldsymbol{\eta} \right) - \dot{\gamma}_{\varphi_{\boldsymbol{\theta}}(\boldsymbol{x}_1), \varphi_{\boldsymbol{\theta}}(\boldsymbol{x}_0)}^{\mathcal{M}}(t) \right\|_{\gamma_{\varphi_{\boldsymbol{\theta}}(\boldsymbol{x}_1), \varphi_{\boldsymbol{\theta}}(\boldsymbol{x}_0)}^{\mathcal{M}}(t)}^{\mathcal{M}} \right)^2, \tag{9}$$

Assuming a latent manifold $\mathcal{M}$ with closed-form mappings enables simulation-free training on data manifolds. For efficiency, we model the $d$-dimensional latent manifold as a product manifold, $\mathcal{M} = \mathcal{M}_{d'} \times \mathbb{R}^{d-d'}$. By encoding samples close to the submanifold $\mathcal{M}_{d'} \subset \mathcal{M}$, isometric learning ensures geodesics $\mathcal{M}_{d'}$ closely match geodesics on $\mathcal{M}$. As a result, we formulate the $d'$-PFM objective,

$$\mathcal{L}_{d'-PFM}(\boldsymbol{\eta}) =$$
$$\mathbb{E}_{t, q(\boldsymbol{x}_1), p(\boldsymbol{x}_0)} \left( \left\| v_t \left( \gamma_{\varphi_{\boldsymbol{\theta}}(\boldsymbol{x}_1), \varphi_{\boldsymbol{\theta}}(\boldsymbol{x}_0)}^{\mathcal{M}_{d'}}(t); \boldsymbol{\eta} \right) - \dot{\gamma}_{\varphi_{\boldsymbol{\theta}}(\boldsymbol{x}_1), \varphi_{\boldsymbol{\theta}}(\boldsymbol{x}_0)}^{\mathcal{M}_{d'}}(t) \right\|_{\gamma_{\varphi_{\boldsymbol{\theta}}(\boldsymbol{x}_1), \varphi_{\boldsymbol{\theta}}(\boldsymbol{x}_0)}^{\mathcal{M}_{d'}}(t)}^{\mathcal{M}_{d'}} \right)^2, \tag{10}$$

The $d'$-PFM objective offers two key benefits. First, defining the objective on the submanifold $\mathcal{M}_{d'}$ results in computational speed-ups during training. Second, the known geometry on the submanifold simplifies the training dynamics of the vector field $v_t(\cdot; \boldsymbol{\eta})$, requiring fewer parameters $\boldsymbol{\eta}$ to learn the sampling trajectories of the data manifold, see Table 3.

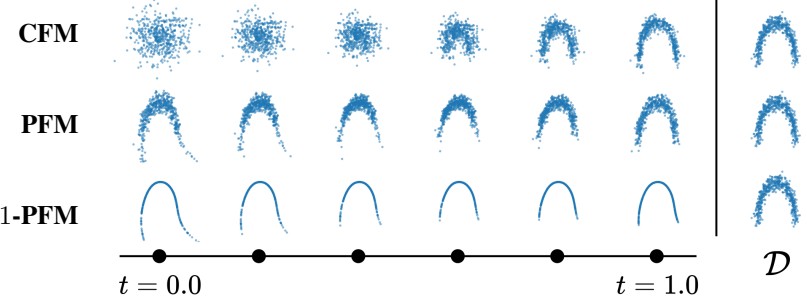

Figure 3: Trajectories of continous normalizing flows (CNF) (left) trained with Conditional Flow Matching (CFM), PFM and 1-PFM objectives on the ARCH dataset compared to the data manifold $\mathcal{D}$ (right). At $t = 0$ the trajectory starts with a standard normal distribution in the data space for CFM and latent submanifold for (1-)PFM mapped back to the data space through $\varphi_{\boldsymbol{\theta}}^{-1}$.

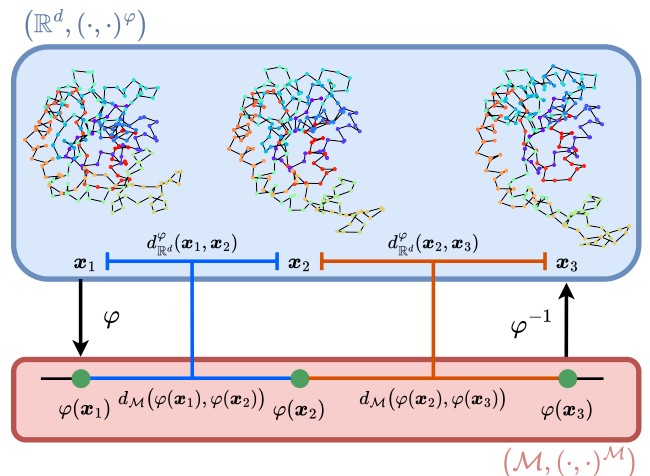

Figure 4: Isometric learning for coarse-grained protein conformation data of adynalate kinase. We define a new metric $(\cdot,\cdot)^\varphi$ on the entire ambient space, $\mathbb{R}^d$ ($d = 214 \times 3$), by learning a diffeomorphism $\varphi : \mathbb{R}^d \to \mathcal{M}$ that preserves a locally Euclidean metric $(\cdot,\cdot)^\mathcal{D}$ on the latent manifold $\mathcal{M} = \mathcal{M}_{d'} \times \mathbb{R}^{d-d'}$ for $d' = 1$.

## 4  LEARNING ISOMETRIES

The motivation for learning isometries $\varphi_{\boldsymbol{\theta}}$—metric-preserving diffeomorphisms—is to enable a latent (sub)manifold that supports interpolation with closed-form geometric mappings, facilitating simulation-free training of PFM. Building on the framework of Diepeveen (2024), summarized in Appendix A, we propose a more expressive parameterization of learnable diffeomorphisms $\varphi_{\boldsymbol{\theta}}$ through Neural ODEs and enhance the objective for scalable isometric learning on data manifolds.

### 4.1  PARAMETERIZING DIFFEOMORPHISMS

We parameterize diffeomorphisms, invertible and differentiable functions between two manifolds, specifically $\varphi : \mathbb{R}^d \to \mathcal{M}$. In practice, we construct the latent manifold as a product manifold, $\mathcal{M} = \mathcal{M}_{d'} \times \mathbb{R}^{d-d'}$ and the diffeomorphism $\varphi$ as,

$$\varphi := [\psi^{-1}, \boldsymbol{I}_{d-d'}] \circ \phi \circ T_{\boldsymbol{\mu}}, \tag{11}$$

where $\psi : U \to \mathbb{R}^{d'}$ a chart on a geodesically convex subset $U \subset \mathcal{M}_{d'}$ of the $d'$-dimensional latent submanifold $(\mathcal{M}_{d'}, (\cdot,\cdot)_{\mathcal{M}_{d'}})$, $\phi : \mathbb{R}^d \to \mathbb{R}^d$ a diffeomorphism and $T_{\boldsymbol{\mu}}(\boldsymbol{x}) = \boldsymbol{x} - \boldsymbol{\mu}$, with $\boldsymbol{\mu}$ the average of the datapoints. We choose this construction because the manifold hypothesis translates to assuming the data manifold is homeomorphic to $\mathcal{M}_{d'}$. In such case, the rest of the latent manifold should be mapped close to zero, e.g. $\varphi(\boldsymbol{x}_i)$ is close to $\mathcal{M}_{d'} \times \boldsymbol{0}^{d-d'}$ in terms of the metric on $\mathcal{M}$.

We generate the diffeomorphism $\phi$ by solving a Neural ODE (Chen et al., 2018). The advantage of this approach is threefold, *i)* this parameterization of diffeomorphisms is more expressive and efficient to train compared to Invertible Residual Networks (Behrmann et al., 2019) as chosen by Diepeveen (2024), *ii)* based on some mild technical assumptions a Neural ODE can be proven to generate proper diffeomorphisms, see Appendix B for the proof, and *iii)* numerically the accuracy and invertibility of the generated flow can be controlled through smaller step-sizes and higher-order solvers.

To define the diffeomorphism $\phi_{\boldsymbol{\theta}} : \mathbb{R}^d \to \mathbb{R}^d$, we start with the Neural ODE governing the flow:

$$\frac{d\boldsymbol{z}(t)}{dt} = f(\boldsymbol{z}(t); \boldsymbol{\theta}), \tag{12}$$

where $f : \mathbb{R}^d \to \mathbb{R}^d$ is a vector field parameterized by a multilayer perceptron (MLP) with Swish activation functions and a sine-cosine time embedding and $\boldsymbol{\theta}$ denotes the parameters of the MLP. Given an initial condition $\boldsymbol{z}(0) = \boldsymbol{x}$, the solution to this Neural ODE is:

$$\phi_{\boldsymbol{\theta}}(\boldsymbol{x}) := \boldsymbol{x} + \int_0^1 f(\boldsymbol{z}(t); \boldsymbol{\theta}) \, dt. \tag{13}$$

To obtain the inverse $\phi_{\boldsymbol{\theta}}^{-1}$ one has to integrate the differential equation backwards in time with initial condition $\boldsymbol{z}(1)$. To solve the Neural ODE we implemented a Runge-Kutta solver in JAX, see Appendix E for further architectural and training related details.

## 4.2 LEARNING OBJECTIVE

The primary objectives of learning isometries are *i)* to map the data manifold $\left(\mathcal{D}, (\cdot, \cdot)^{\mathcal{D}}\right)$ into a low-dimensional geodesic subspace of $\left(\mathcal{M}, (\cdot, \cdot)^{\mathcal{M}}\right)$, specifically $\mathcal{M}_{d'} \subset \mathcal{M}$, and *ii)* to preserve local isometry, as motivated by Proposition 2.1 and Theorems 3.4, 3.6, and 3.8 from Diepeveen (2024).

We build on the training objective from Diepeveen (2024) (summarized in Appendix A) and use **global isometry loss** and **submanifold loss** to map the data manifold $\mathcal{D}$ to the lower-dimensional geodesic subspace $\mathcal{M}_{d'}$. We enhance this with the **graph matching loss** for isometric learning, which enforces global isometry between the data and latent manifolds (Zhu et al., 2014), ensuring that each sample is equally isometric to all others.

The original objective enforces local isometry—preserving geodesic distances in small neighborhoods—via the pullback metric's Riemannian tensor $(\cdot, \cdot)^{\varphi}$. However, this is computationally intractable and poorly scalable. We address this by using the regularization in **stability regularization** from Finlay et al. (2020), which more efficiently enforces local isometry, leading to a scalable objective,

$$
\mathcal{L}(\boldsymbol{\theta}) = \alpha_1 \frac{1}{n^2} \sum_{i=1}^{n} \sum_{j=1}^{n} \| d_{\mathbb{R}^d}^{\varphi_{\boldsymbol{\theta}}}(\boldsymbol{x}_i, \boldsymbol{x}_j) - d_{i,j} \|^2 \qquad \text{(\textbf{global isometry loss})}
$$

$$
+ \alpha_2 \frac{1}{n} \sum_{i=1}^{n} \sum_{j \neq i} \| (\boldsymbol{d}_{\mathbb{R}^d}^{\varphi_{\boldsymbol{\theta}}}(\boldsymbol{x}_i, \boldsymbol{x}_{\cdot}) - \boldsymbol{d}_{\mathbb{R}^d}^{\varphi_{\boldsymbol{\theta}}}(\boldsymbol{x}_j, \boldsymbol{x}_{\cdot})) - (\boldsymbol{d}_{i,\cdot} - \boldsymbol{d}_{j,\cdot}) \|^2 \quad \text{(\textbf{graph matching loss})}
$$

$$
+ \alpha_3 \frac{1}{n} \sum_{i=1}^{n} \left\| \begin{bmatrix} \mathbf{0}_{d'} & \emptyset \\ \emptyset & \boldsymbol{I}_{d-d'} \end{bmatrix} (\phi_{\boldsymbol{\theta}} \circ T_{\boldsymbol{\mu}})(\boldsymbol{x}_i) \right\|_1 \qquad \text{(\textbf{submanifold loss})}
$$

$$
+ \alpha_4 \frac{1}{n} \sum_{i=1}^{n} \int_0^1 \| \boldsymbol{\varepsilon}^T \nabla f_{\boldsymbol{\theta}}(\boldsymbol{z}_i(t)) \|^2 \, dt. \qquad \text{(\textbf{stability regularization})}
$$

Here, $\boldsymbol{\varepsilon} \sim \mathcal{N}(\mathbf{0}, \boldsymbol{I})$ and $\boldsymbol{d}_{\mathbb{R}^d}^{\varphi_{\boldsymbol{\theta}}}(\boldsymbol{x}_i, \boldsymbol{x}_{\cdot})$ and $\boldsymbol{d}_{i,\cdot}$ denote the columns of the distance matrices induced by $(\cdot, \cdot)^{\varphi}$ and $(\cdot, \cdot)^{\mathcal{D}}$. The benefit of this formulation is that it only requires approximating geodesic distances $d_{i,j}$ on the data manifold $\mathcal{D}$, without needing to calculate or differentiate the metric tensor. In section 5, we demonstrate the effectiveness of the graph matching loss and stability regularization through an ablation study on synthetic and high-dimensional protein dynamics trajectories. We do not include an ablation of the global isometry loss and submanifold losses, as these have been thoroughly examined in Diepeveen (2024), and our experiments showed consistent results with those previously reported.

## 5 EXPERIMENTS

The goal of this paper is to learn interpolatable latent (sub)manifolds for generation on data manifolds. We achieve this through isometric learning in the framework of pullback geometry. In this section we validate our methods on synthetic, simulated and experimental datasets, for full descriptions see Appendix D. For details on the training procedure and hyperparameter settings we refer the reader to Appendix E.

We begin our experiments with an ablation study of **graph matching loss** and **stability regularization**, demonstrating the benefits of including both terms for learning isometries. Second, we compare (latent) interpolation methods with interpolation on the latent manifold $\mathcal{M}$, $(\cdot, \cdot)^{\mathcal{M}}$-interpolation, and on the latent submanifold $\mathcal{M}_{d'}$, $(\cdot, \cdot)^{\mathcal{M}_{d'}}$-interpolation. We demonstrate that we can accurately interpolate on the data manifold by interpolating on the latent (sub)manifold [2]. Third, we validate PFM as a generative model on data manifolds and discuss how sample generation is improved by generating on the submanifold $\mathcal{M}_{d'}$. Finally, we inspect the designability of the latent manifold through the choice of metric $(\cdot, \cdot)^{\mathcal{D}}$ in the task of small protein design.

---

[2]In these experiments we do not report interpolation through the Riemannian Auto-Encoder (RAE) by Diepeveen (2024) due to the intractability of the training objective for the higher-dimensional datasets.

## 5.1 ABLATION STUDY

The goal of the ablation study is to evaluate the effectiveness of the reformulated objective function for learning isometries. To this end, we perform an ablation study for both the **graph matching loss** and **stability regularization** on a synthetic ARCH dataset ($n = 500$, $d = 2$) in the spirit of Tong et al. (2020) and a coarse-grained protein dynamics datasets of intestinal fatty acid binding protein (I-FABP) ($n = 500$, $d = 131 \times 3$). We report three metrics on the validation set of 20 % of the data, invertibility $\varepsilon_{inv} = \frac{1}{n} \sum_{i=1}^{n} \| \boldsymbol{x}_i - \varphi_{\boldsymbol{\theta}}^{-1}(\varphi_{\boldsymbol{\theta}}(\boldsymbol{x}_i)) \|^2$, low-dimensionality $\varepsilon_{ld} = \frac{1}{n} \sum_{i=1}^{n} \left\| \begin{bmatrix} \mathbf{0}_{d'} & \emptyset \\ \emptyset & \boldsymbol{I}_{d-d'} \end{bmatrix} \phi_{\boldsymbol{\theta}}(\boldsymbol{x}_i) \right\|_1^2$ and isometry $\varepsilon_{iso} = \frac{1}{n^2} \sum_{i=1}^{n} \sum_{j=1}^{n} \| d_{i,j} - d_{\mathcal{M}}(\varphi(\boldsymbol{x}_i), \varphi(\boldsymbol{x}_j)) \|^2$.

Table 1: Ablation study of isometric learning for ARCH dataset and I-FABP protein dynamics datasets for **graph matching loss** (GM) and **stability regularization** (Stability). In both cases we choose $\mathcal{M}_{d'} = \mathbb{R}$. We report the means for invertibility ($\downarrow$), low-dimensionality ($\downarrow$) and isometry ($\downarrow$) with standard devations denoted by $\pm$. The distance $(\cdot, \cdot)^{\mathcal{D}}$ we assume on the data manifold $\mathcal{D}$ is a locally Euclidean distance based on Isomap (Tenenbaum et al., 2000).

| Data | Metric | None | GM | Stability | Both |
|---|---|---|---|---|---|
| **ARCH** | Invertibility | $7.637 \cdot 10^{-1}$ $\pm 9.872 \cdot 10^{-1}$ | $3.585 \cdot 10^{-2}$ $\pm 1.939 \cdot 10^{-2}$ | $\mathbf{8.198 \cdot 10^{-5}}$ $\pm 1.061 \cdot 10^{-5}$ | $1.011 \cdot 10^{-4}$ $\pm 6.069 \cdot 10^{-5}$ |
| | Low-Dimensionality | $6.520 \cdot 10^{-4}$ $\pm 9.521 \cdot 10^{-5}$ | $\mathbf{4.531 \cdot 10^{-4}}$ $\pm 3.341 \cdot 10^{-5}$ | $1.407 \cdot 10^{-2}$ $\pm 8.414 \cdot 10^{-4}$ | $1.373 \cdot 10^{-2}$ $\pm 6.768 \cdot 10^{-4}$ |
| | Isometry | $2.334 \cdot 10^{-3}$ $\pm 1.466 \cdot 10^{-4}$ | $\mathbf{1.464 \cdot 10^{-3}}$ $\pm 1.221 \cdot 10^{-4}$ | $2.018 \cdot 10^{-3}$ $\pm 5.791 \cdot 10^{-5}$ | $1.544 \cdot 10^{-3}$ $\pm 2.025 \cdot 10^{-4}$ |
| **I-FABP** | Invertibility | $2.995 \cdot 10^{-5}$ $\pm 8.945 \cdot 10^{-6}$ | $2.891 \cdot 10^{-5}$ $\pm 4.968 \cdot 10^{-6}$ | $2.973 \cdot 10^{-5}$ $\pm 7.560 \cdot 10^{-6}$ | $\mathbf{2.809 \cdot 10^{-5}}$ $\pm 8.982 \cdot 10^{-6}$ |
| | Low-Dimensionality | $\mathbf{1.378 \cdot 10^{-1}}$ $\pm 1.952 \cdot 10^{-4}$ | $1.379 \cdot 10^{-1}$ $\pm 1.424 \cdot 10^{-4}$ | $1.379 \cdot 10^{-1}$ $\pm 1.788 \cdot 10^{-4}$ | $\mathbf{1.378 \cdot 10^{-1}}$ $\pm 1.981 \cdot 10^{-4}$ |
| | Isometry | $2.889 \cdot 10^{-3}$ $\pm 1.384 \cdot 10^{-4}$ | $2.898 \cdot 10^{-3}$ $\pm 1.667 \cdot 10^{-4}$ | $2.919 \cdot 10^{-3}$ $\pm 1.571 \cdot 10^{-4}$ | $\mathbf{2.887 \cdot 10^{-3}}$ $\pm 1.387 \cdot 10^{-4}$ |

**Result.** Table 1 demonstrates that incorporating both the graph matching loss and stability regularization improves the invertibility and isometry metrics across both datasets, with the combined approach yielding both a low $\varepsilon_{inv}$ and $\varepsilon_{iso}$ values, indicating enhanced model performance in preserving the geometry of the data in the synthetic dataset as well as the more noisy and high dimensional simulated dataset.

## 5.2 INTERPOLATION EXPERIMENTS

The goal of isometric learning is to learn an interpolatable latent (sub)manifold of the data manifold with closed-form manifold mappings. To evaluate whether interpolation on the latent (sub)manifold accurately reflects interpolation on the data manifold, we conduct an interpolation experiment using the synthetic ARCH dataset, as well as the molecular dynamics datasets of Adenylate Kinase (AK) ($n = 100$, $d = 214 \times 3$) and I-FABP.

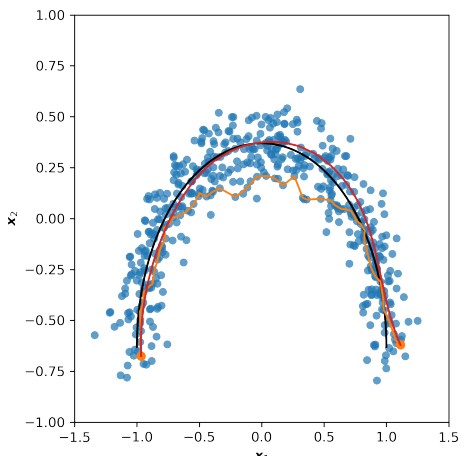

Figure 5: Example of $(\cdot, \cdot)^{\mathcal{M}_{d'}}$-interpolation for ARCH dataset in **red**. In **blue** the dataset $\{\boldsymbol{x}_i\}_{i=1}^{n}$, **black** the true submanifold $\mathcal{M}_{d'}$, the half circle, and in **orange** the Isomap geodesic between **orange points**.

In both cases we choose $\mathcal{M}_{d'} = \mathbb{R}$, see Appendix C for guidance on latent manifold and metric selection. We approximate the metric on the data manifold $(\cdot, \cdot)^{\mathcal{D}}$ through the length of Isomap's geodesics Tenenbaum et al. (2000), see Figure 5 for an example. We compare the accuracy of the 100 longest geodesics between points in the test set for multiple (latent) interpolation methods.

Table 2: Root mean square error (RMSE) ($\downarrow$) of the 100 longest isomap geodesics between points in the test set for 3 different seeds for different latent interpolation methods. We compare our methods, $(\cdot, \cdot)^{\mathcal{M}}$-interpolation and $(\cdot, \cdot)^{\mathcal{M}_{d'}}$-interpolation, with variational autoencoders (VAEs) (Kingma & Welling, 2013), $\beta-$VAEs (Higgins et al., 2017) and GRAE (Duque et al., 2022).

| Interpolation | Latent | ARCH | Swiss Roll | AK | I-FABP |
|---|---|---|---|---|---|
| Linear | ✗ | $0.331_{\pm 0.049}$ | $0.573_{\pm 0.018}$ | $0.554_{\pm 0.131}$ | $0.494_{\pm 0.022}$ |
| VAE | ✓ | $0.526_{\pm 0.024}$ | $0.596_{\pm 0.085}$ | $1.235_{\pm 0.477}$ | $0.405_{\pm 0.023}$ |
| $\beta$-VAE | ✓ | $0.527_{\pm 0.025}$ | $0.640_{\pm 0.066}$ | $0.919_{\pm 0.631}$ | $0.368_{\pm 0.009}$ |
| GRAE (Isomap) | ✓ | $0.426_{\pm 0.076}$ | $0.568_{\pm 0.024}$ | $2.030_{\pm 0.579}$ | $0.442_{\pm 0.005}$ |
| GRAE (PHATE) | ✓ | $0.128_{\pm 0.052}$ | $0.660_{\pm 0.150}$ | $1.012_{\pm 0.395}$ | $0.474_{\pm 0.040}$ |
| $(\cdot, \cdot)^{\mathcal{M}}$ | ✓ | $\mathbf{0.097}_{\pm 0.030}$ | $\mathbf{0.159}_{\pm 0.054}$ | $0.296_{\pm 0.058}$ | $0.415_{\pm 0.025}$ |
| $(\cdot, \cdot)^{\mathcal{M}_{d'}}$ | ✓ | $0.109_{\pm 0.026}$ | $0.159_{\pm 0.055}$ | $\mathbf{0.219}_{\pm 0.012}$ | $\mathbf{0.292}_{\pm 0.006}$ |

**Result.** The $(\cdot, \cdot)^{\mathcal{M}_{d'}}$- and $(\cdot, \cdot)^{\mathcal{M}}$-interpolation achieves superior interpolation accuracy with lower root mean square error (RMSE) variablity compared to other models, indicating more robust and reliable interpolation. $(\cdot, \cdot)^{\mathcal{M}_{d'}}$-interpolation specifically demonstrates improvements over other methods in the more stochastic and seemingly higher dimensional AK ($d = 639$) and I-FABP ($d = 642$) datasets. This improvement suggests that compressing the latent representation into a lower-dimensional space reduces the noise while accurately capturing the underlying data manifold. Our findings demonstrate that accurate interpolation of protein dynamics trajectories of AK and I-FABP can be achieved using a single-dimensional latent manifold. This method shows promise for improving protein dynamics simulations, ultimately advancing understanding of protein dynamics.

## 5.3 Generation Experiments

We demonstrate the effectiveness of our proposed method PFM for generation on data manifolds $\mathcal{D}$. We train two PFMs, one using the latent manifold $\mathcal{M}$ and one using the lower dimensional latent submanifold $\mathcal{M}_{d'}$, named PFM and $d'$-PFM respectively. Additionally, we train a Conditional Flow Matching (CFM) model on the raw data as a comparison. A visual example of the learned generative flows over time for the ARCH dataset can be viewed in Figure 3. To evaluate our generative methods, we use the 1-nearest neighbour (NN) accuracy (Lopez-Paz & Oquab, 2016), which measures how well the generated point clouds match the reference point clouds. Each point cloud is classified by finding its nearest neighbor in the combined set of generated and reference point clouds. The accuracy reflects how similar the generated point clouds are to the reference set, with an accuracy close to $50\%$ indicating successful learning of the target distribution.

Table 3: Evaluation of generative model performance across dimensionality of (latent) (sub)manifold ($\downarrow$), number of model parameters, denoted by # pars ($\downarrow$), and 1-NN accuracy (1-NN$\rightarrow 0.5$). The 1-NN metric measures the generative quality, with values closer to $0.5$ indicating better performance.

| | | ARCH | | | Swiss | |
|---|---|---|---|---|---|---|
| **Model** | dim | # pars | **1-NN** | dim | # pars | **1-NN** |
| CFM | 2 | 50562 | $0.295_{\pm 0.031}$ | 2 | 50691 | $0.870_{\pm 0.016}$ |
| PFM | 2 | 50562 | $0.262_{\pm 0.025}$ | 2 | 50691 | $0.795_{\pm 0.011}$ |
| 1-PFM | 1 | **5697** | $\mathbf{0.487}_{\pm 0.027}$ | 1 | **16066** | $\mathbf{0.789}_{\pm 0.019}$ |
| | | AK | | | I-FABP | |
| **Model** | dim | # pars | **1-NN** | dim | # pars | **1-NN** |
| CFM | 642 | 4682325 | $0.386_{\pm 0.000}$ | 393 | 1789941 | $0.365_{\pm 0.004}$ |
| PFM | 642 | 4682325 | $0.356_{\pm 0.097}$ | 393 | 1789941 | $0.452_{\pm 0.017}$ |
| 1-PFM | 1 | **5697** | $\mathbf{0.464}_{\pm 0.022}$ | 1 | **5697** | $\mathbf{0.508}_{\pm 0.006}$ |

**Result.** Figure 3 we see that the learned isometry to the latent manifold $\mathcal{M}$ acts as a strong manifold prior, capturing the manifold structure at the start of the continous normalizing flows (CNF) trajectory ($t = 0.0$). Additionally, the learned isometry to the latent submanifold $\mathcal{M}_{d'}$ captures

the noiseless manifold revealing the underlying manifold used to generate the data. Through this strong (noiseless) manifold prior, we see that both PFM and 1-PFM approximate the distribution on the manifold earlier in the trajectory and better. Table 3 highlights the effectiveness of the 1-PFM model in generative tasks. The 1-PFM model leverages the lower-dimensional isometric latent manifold $\mathcal{M}_{d'}$, significantly reducing the number of parameters required. Training the 1-PFM is significantly faster due to the reduction in parameters and the dimensionality of the training samples. The 1-NN accuracy for 1-PFM approaches the ideal $0.5$ across all datasets, indicating that this model better captures the underlying distribution on the data manifold compared to CFM and PFM.

## 5.4 DESIGNABLE LATENT MANIFOLDS FOR NOVEL PROTEIN ENGINEERING

The goal of these experiments is to design a latent manifold that captures biologically relevant properties of protein sequences, enabling the generation of novel proteins with specific characteristics. By leveraging our method's flexibility in defining the metric on the data manifold $(\cdot, \cdot)^{\mathcal{D}}$, we structure the latent space such that it captures protein properties, such as sequence similarity, hydrophobicity, hydrophobic moment, charge, and isoelectric point.

To achieve this, we use protein sequences of up to 25 amino acids from the giant repository of AMP activities (GRAMPA) dataset (see Appendix D for details). We construct the following custom metric on the data manifold,

$$d_{\mathcal{D}}(x_i, x_j) = d_{\text{Levenshtein}}(x_i, x_j) + d_{\text{hydrophobicity}}(x_i, x_j) \tag{14}$$

$$+ d_{\text{hydrophobic moment}}(x_i, x_j) + d_{\text{charge}}(x_i, x_j) \tag{15}$$

$$+ d_{\text{isoelectric point}}(x_i, x_j), \tag{16}$$

where the Levenshtein distance measures the number of single-character edits (insertions, deletions, or substitutions) required to transform one sequence into another.

For the remaining four properties—hydrophobicity, hydrophobic moment, charge, and isoelectric point—distances are computed using the difference in property values between sequences. Specifically, for each property, we define the (pseudo)distance as,

$$d_{[\text{property}]}(x_i, x_j) = |f_{\text{property}}(x_i) - f_{\text{property}}(x_j)|. \tag{17}$$

These (pseudo)distances are standardized by dividing by the maximum observed distance in the training data. Since the Levenshtein distance is a proper metric, we ensure that the combined distance $d_{\mathcal{D}}(x_i, x_j)$ remains a valid distance metric.

We use the designed metric $(\cdot, \cdot)^{\mathcal{D}}$ on the space of protein sequences with at most 25 amino acids in the GRAMPA dataset to learn an isometry that preserves this metric on the latent manifold $\mathcal{M}$ and latent submanifold $\mathcal{M}_{d'}$.

To generate protein sequences with specific properties, we sample from a normal distribution around the data points in the latent manifold $\boldsymbol{z} \in \mathcal{M}$ or latent submanifold $\boldsymbol{z} \in \mathcal{M}_{d'}$. The variability of this sampling process is aligned with the latent variability of the training data $\sigma_{\boldsymbol{z}_{train}}$, scaled by a temperature factor $\tau$, resulting in the following expression,

$$\boldsymbol{z}_i^{(analogue)} = \boldsymbol{z}_i + \tau \mathcal{N}(\boldsymbol{0}, \sigma_{\boldsymbol{z}_{train}} \boldsymbol{I}), \text{ and} \tag{18}$$

$$\boldsymbol{x}_i^{(analogue)} = \varphi_{\boldsymbol{\theta}}^{-1}(\boldsymbol{z}_i) \text{ for } i = 1, \dots, n_{test}. \tag{19}$$

This sampling methodology is referred to as *analogue generation*, as it does not involve explicitly learning the distribution over the latent manifold. Instead, it generates novel sequences by sampling around existing data points on the latent (sub)manifold of the test set.

We apply this process to both the latent manifold $\mathcal{M}$ and its submanifold $\mathcal{M}_{d'}$. To evaluate the effectiveness of the generated sequences, we measure the number of unique sequences that were not present in the original dataset and compare the properties of the generated samples to the properties of their base points. For further specifics on hyperparameters and training procedures, refer to Appendix E.

**Results.** The application of our designed latent manifold facilitated the generation of diverse novel protein sequences, demonstrating the effectiveness of the analogue generation methodology. As illustrated in Table 4, increasing the temperature parameter, $\tau$, directly influenced the diversity of generated sequences. At lower temperatures ($\tau \leq 0.1$), many unique sequences emerged while maintaining similarity to their base points, as indicated by non-significant KS test values. Conversely,

Table 4: Unique protein sequences generated via analogue generation on the latent manifold $\mathcal{M}$ and its submanifold $\mathcal{M}_{d'}$ at various temperatures ($\tau$). The table presents the total sequences generated (Total), those already in the dataset (In Data), and the number of novel sequences (Novel). We perform a Kolmogorov-Smirnov test at a $5\%$ significance level to compare novel sequences with their base points. Non-significant Kolmogorov-Smirnov values are shown as X/Y, where X is the number of non-significant properties and Y is the total properties tested.

| | $\mathcal{M}$ | | | | $\mathcal{M}_{d'}$ | | | |
|---|---|---|---|---|---|---|---|---|
| $\tau$ | **Total** | **In Data** | **Novel** | **Non-Sign. KS** | **Total** | **In Data** | **Novel** | **Non-Sign. KS** |
| 0.01 | 689 | 652 | 37 | 5/5 | 687 | 5 | 682 | 2/5 |
| 0.05 | 689 | 103 | 586 | 5/5 | 689 | 4 | 685 | 2/5 |
| 0.1 | 689 | 35 | 654 | 5/5 | 689 | 4 | 685 | 2/5 |
| 0.2 | 689 | 12 | 677 | 2/5 | 689 | 0 | 689 | 2/5 |
| 0.5 | 689 | 1 | 688 | 1/5 | 689 | 0 | 689 | 1/5 |
| 1 | 689 | 0 | 689 | 0/5 | 689 | 0 | 689 | 0/5 |

higher temperatures ($\tau > 0.1$) resulted in a significant increase in novel sequences, accompanied by significant KS values suggesting greater divergence from base sequences. This observation supports the hypothesis that novel sequences generated close to the base points are structurally similar, highlighting the effectiveness of isometric learning in structuring the latent space. Overall, our results indicate that temperature manipulation can strategically balance novelty and similarity, paving the way for innovative applications in protein engineering.

Latent interpolation experiments, illustrated in Figure 6, further demonstrate the potential of our approach. By interpolating between sequences with contrasting properties, we revealed a smooth transition of characteristics within the latent space, reinforcing our method's capability to fine-tune specific protein attributes. This smooth transition indicates that our latent manifold can be effectively navigated to explore a continuum of properties such as hydrophobicity, hydrophobic moment, charge, and isoelectric point, which are essential for determining protein solubility, stability, and interaction behavior. This capability allows for the targeted design of protein sequences that could be optimized for specific biochemical contexts, potentially enhancing their performance in applications like enzyme catalysis or therapeutic development. In summary, the efficacy of our designed latent manifold not only expands the repertoire of available protein sequences but

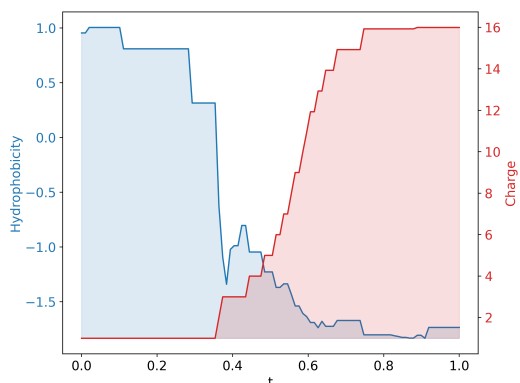

Figure 6: Latent interpolation between a protein with a high hydrophobic moment and low charge and a protein with a low hydrophobic moment and high charge.

also ensures retention of biologically relevant properties, positioning this approach as a valuable tool for precision in protein engineering.

## 6 CONCLUSION

We introduce Pullback Flow Matching (PFM), a novel framework for simulation-free training of generative models on data manifolds. By leveraging pullback geometry and isometric learning, PFM allows for closed-form mappings on data manifolds while enabling precise interpolation and efficient generation. We demonstrated the effectiveness of PFM through applications in synthetic protein dynamics and small protein generation, showcasing its potential in generating novel, property-specific samples through designable latent spaces. This approach holds significant promise for advancing generative modeling in fields like drug discovery and materials science, where precise and efficient sample generation is critical.

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

# A  BACKGROUND

To achieve an interpolatable latent manifold we take a Riemannian geometric perspective. We start by introducing the notation and key concepts of differential and Riemannian geometry, for a formal description see Lee (2012). Second, we explain prior work on RAEs Diepeveen (2024), a framework for constructing interpolatable latent manifolds. Third, we summarize CFM for generative modeling Lipman et al. (2022), a scalable way to train generative models in a simulation-free manner. Finally, we discuss how RFM Chen & Lipman (2024) generalize CFM to Riemannian manifolds.

## A.1  RIEMMANIAN GEOMETRY

A $d$-dimensional *smooth manifold* $\mathcal{M}$ is a topological space that locally resembles $\mathbb{R}^d$, such that for each point $\boldsymbol{p} \in \mathcal{M}$, there exists a neighborhood $U$ of $\boldsymbol{p}$ and a *homeomorphism* $\psi : U \to \mathbb{R}^d$, called a *chart*. Then the *tangent space* $\mathcal{T}_{\boldsymbol{p}}\mathcal{M}$ at a point $\boldsymbol{p} \in \mathcal{M}$ is a vector space consisting of the tangent vectors at $\boldsymbol{p}$ representing the space of derivations at $\boldsymbol{p}$.

A *Riemannian manifold* $\left(\mathcal{M}, (\cdot, \cdot)^{\mathcal{M}}\right)$ is a smooth manifold $\mathcal{M}$ equipped with a *Riemannian metric* $(\cdot, \cdot)^{\mathcal{M}}$, which is a smoothly varying positive-definite inner product on the tangent space $\mathcal{T}_{\boldsymbol{p}}\mathcal{M}$ at each point $\boldsymbol{p}$. The Riemannian metric $(\cdot, \cdot)^{\mathcal{M}}$ defines the length of tangent vectors and the angle between them, thereby inducing a natural notion of distance on $\mathcal{M}$ based on the lengths of tangent vectors along curves between two points.

The shortest path between two points on $\mathcal{M}$ is called a *geodesic*, which generalizes the concept of straight lines in Euclidean space to curved manifolds. Geodesics on Riemannian manifold are found by minimizing

$$E(\gamma) = \frac{1}{2} \int_0^1 \left(\dot{\gamma}(t), \dot{\gamma}(t)\right)_{\gamma(t)} dt, \tag{20}$$

whereas

$$L(\gamma) = \int_0^1 \sqrt{\left(\dot{\gamma}(t), \dot{\gamma}(t)\right)_{\gamma(t)}} dt \tag{21}$$

defines the distance between two points on the manifold. The *exponential map*,

$$\exp_{\boldsymbol{p}} : \mathcal{T}_{\boldsymbol{p}}\mathcal{M} \to \mathcal{M}, \tag{22}$$

at $\boldsymbol{p}$ maps a tangent vector $\Xi_{\boldsymbol{p}} \in \mathcal{T}_{\boldsymbol{p}}\mathcal{M}$ to a point on $\mathcal{M}$ reached by traveling along the geodesic starting at $\boldsymbol{p}$ in the direction of $\Xi_{\boldsymbol{p}}$ for unit time. The *logarithmic map*,

$$\log_{\boldsymbol{p}} : \mathcal{M} \to \mathcal{T}_{\boldsymbol{p}}\mathcal{M}, \tag{23}$$

is the inverse of the exponential map, mapping a point $\boldsymbol{q} \in \mathcal{M}$ back to the tangent space $\mathcal{T}_{\boldsymbol{p}}\mathcal{M}$ at $\boldsymbol{p}$.

These names, 'exponential' and 'logarithmic' map, are geometric extensions of familiar calculus concepts. Just as the exponential function maps a number to a point on a curve, the exponential map on a manifold maps a direction and starting point to a location along a geodesic. Similarly, the logarithm in calculus reverses exponentiation, and the logarithmic map on a manifold reverses the exponential map, returning the original direction and distance needed to reach a specified point along the geodesic.

Assume $\left(\mathcal{M}, (\cdot, \cdot)^{\mathcal{M}}\right)$ is a $d$-dimensional Riemannian manifold and a smooth diffeomorphism $\varphi : \mathbb{R}^d \to \mathcal{M}$, such that $\varphi(\mathbb{R}^d) \subseteq \mathcal{M}$ is geodesically convex, i.e., geodesics are uniquely defined on $\varphi(\mathbb{R}^d)$. We can then define the *pullback metric* as

$$(\Xi_{\mathbf{p}}, \Phi_{\mathbf{p}})_{\mathbf{p}}^{\varphi} := \left(\varphi_*[\Xi_{\mathbf{p}}], \varphi_*[\Phi_{\mathbf{p}}]\right)_{\varphi(\mathbf{p})}, \tag{24}$$

for tangent vectors $\Xi_{\mathbf{p}}$ and $\Phi_{\mathbf{p}}$, where $\varphi_*$ is the pushforward. These mappings allow us to define all relevant geometric mappings in $\mathbb{R}^d$ in terms of manifold mappings on $\mathcal{M}$, see e.g. Proposition 2.1 of Diepeveen (2024):

1. Distances $d_{\mathbb{R}^d}^{\varphi} : \mathbb{R}^d \times \mathbb{R}^d \to \mathbb{R}$ on $\left(\mathbb{R}^d, (\cdot, \cdot)^{\varphi}\right)$ are given by,

$$d_{\mathbb{R}^d}^{\varphi}(\boldsymbol{x_i}, \boldsymbol{x_j}) = d_{\mathcal{M}}\left(\varphi(\boldsymbol{x_i}), \varphi(\boldsymbol{x_j})\right), \tag{25}$$

2. Length-minimizing geodesics $\gamma^{\varphi}_{\boldsymbol{x}_i, \boldsymbol{x}_j} : [0, 1] \rightarrow \mathbb{R}^d$ on $\left( \mathbb{R}^d, (\cdot, \cdot)^{\varphi} \right)$ are given by,

$$\gamma^{\varphi}_{\boldsymbol{x}_i, \boldsymbol{x}_j}(t) = \varphi^{-1} \left( \gamma^{\mathcal{M}}_{\varphi(\boldsymbol{x}_i), \varphi(\boldsymbol{x}_j)}(t) \right) \tag{26}$$

3. Logarithmic maps $\log^{\varphi}_{\boldsymbol{x}_i} : \mathbb{R}^d \rightarrow \mathcal{T}_{\boldsymbol{x}_i} \mathbb{R}^d$ on $\left( \mathbb{R}^d, (\cdot, \cdot)^{\varphi} \right)$ are given by,

$$\log^{\varphi}_{\boldsymbol{x}_i}(\boldsymbol{x}_j) = \varphi^{-1}_* \left[ \log^{\mathcal{M}}_{\varphi(\boldsymbol{x}_i)} \left( \varphi(\boldsymbol{x}_j) \right) \right] \tag{27}$$

4. Exponential maps $\exp^{\varphi}_{\boldsymbol{x}_i} : \mathcal{G}_{\boldsymbol{x}_i} \rightarrow \mathbb{R}^d$ for $\mathcal{G}_{\boldsymbol{x}_i} := \log^{\varphi}_{\boldsymbol{x}_i}(\mathbb{R}^d) \subset \mathcal{T}_{\boldsymbol{x}_i} \mathbb{R}^d$ on $\left( \mathbb{R}^d, (\cdot, \cdot)^{\varphi} \right)$ are given by

$$\exp^{\varphi}_{\boldsymbol{x}_i}(\Xi_{\boldsymbol{x}_i}) = \varphi^{-1} \left( \exp^{\mathcal{M}}_{\varphi(\boldsymbol{x}_i)} \left( \varphi_*[\Xi_{\boldsymbol{x}_i}] \right) \right) \tag{28}$$

A visual example of pullback geometry is given in Figure 7. Pullback geometry allows us to remetrize all of space $\mathbb{R}^d$, including the data manifold $\mathcal{D} \subset \mathbb{R}^d$, through the pullback metric. We can use it to define geometric mappings on $\left( \mathbb{R}^d, (\cdot, \cdot)^{\varphi} \right)$, including geodesics (see Equation 26), through geometric mappings on the latent manifold $\mathcal{M}$. Next, we summarize work on Riemannian Auto-Encoders, that leverage pullback geometry to create an interpolatable latent manifold.

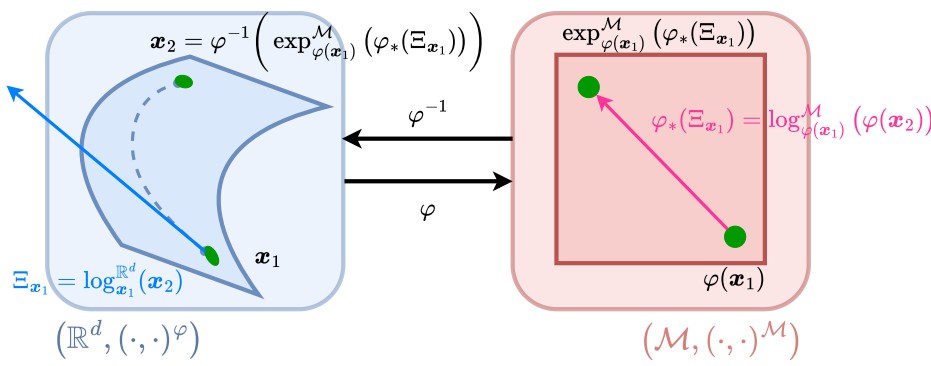

Figure 7: Example of pullback geometry for $\varphi : \mathbb{R}^d \rightarrow \mathcal{M}$ with $\mathcal{M} = \mathcal{M}_{d'} \times \mathbb{R}^{d-d'}$ for $\mathcal{M}_{d'} = \mathbb{R}^{d'}$, $d = 3$ and $d' = 2$. Samples $\varphi(\boldsymbol{x}_i)$ are close to elements of $\mathcal{M}_{d'} \times \mathbf{0}^{d-d'}$.

## A.2 RIEMANNIAN AUTO-ENCODER

The goal of RAEs is to create a interpolatable latent representation of the data. This is achieved through data-driven (pullback) Riemannian geometry, encoding the data onto a latent manifold with known geometry. The benefit of this, is that interpolation on the data manifold corresponds to interpolation on the latent manifold. Resulting in a more interpretable latent space compared to traditional auto-encoders.

Similar as in Diepeveen (2024), we define a RAE as a Riemannian Encoder $RE : \mathbb{R}^d \rightarrow \mathbb{R}^r$ and Riemannian Decoder $RD : \mathbb{R}^r \rightarrow \mathbb{R}^d$,

$$\text{RAE}(\boldsymbol{x}) := (RD \circ RE)(\boldsymbol{x}) \quad s.t., \tag{29}$$

$$\text{RE}(\boldsymbol{x})_k := (\log^{\varphi}_{\boldsymbol{z}}(\boldsymbol{x}), \boldsymbol{v}^k_{\boldsymbol{z}})^{\varphi}_{\boldsymbol{z}} \text{ for } k = 1, \dots r, \tag{30}$$

$$\text{RD}(\boldsymbol{a}) := \exp^{\varphi}_{\boldsymbol{z}} \left( \sum_{k=1}^{r} \boldsymbol{a}_k \boldsymbol{v}^k_{\boldsymbol{z}} \right) \tag{31}$$

where $\boldsymbol{z}$ denotes a base point and $(\cdot, \cdot)^{\varphi}_{\boldsymbol{z}}$ the pullback metric at $\boldsymbol{z}$. Furthermore,

$$\boldsymbol{v}^k_{\boldsymbol{z}} := \sum_{l=1}^{d} \boldsymbol{W}_{lk} \Phi^l_{\boldsymbol{z}}, \tag{32}$$

represents the basis vectors of the latent space in the tangent space $T_{\boldsymbol{z}} \mathbb{R}^d$. Let $\Phi^l_{\boldsymbol{z}} \in T_{\boldsymbol{z}} \mathbb{R}^d$ be an orthonormal basis in the tangent space at $\boldsymbol{z}$ with respect to $(\cdot, \cdot)^{\varphi}_{\boldsymbol{z}}$ and define

$$\boldsymbol{X}_{i,l} = \left( \log^{\varphi}_{\boldsymbol{z}}(\boldsymbol{x}^i), \Phi^l_{\boldsymbol{z}} \right)^{\varphi}_{\boldsymbol{z}} \text{ for } i = 1, \dots, n \text{ and } l = 1, \dots, d. \tag{33}$$

We can compute $\boldsymbol{W}$ through a Singular Value Decomposition (SVD) of $\boldsymbol{X}$.

$$\boldsymbol{X} = \boldsymbol{U}\boldsymbol{\Sigma}\boldsymbol{W}^T, \tag{34}$$

where $U \in \mathbb{R}^{N \times R}$, $\Sigma = \mathrm{diag}(\sigma_1, \ldots, \sigma_R) \in \mathbb{R}^{R \times R}$ with $\sigma_1 \geq \cdots \geq \sigma_R$, $W \in \mathbb{R}^{d \times R}$ and where $R := \mathrm{rank}(\boldsymbol{X})$. The first $r$ columns of $\boldsymbol{W}$, corresponding to the largest singular values, are selected to form the matrix $\boldsymbol{W} \in \mathbb{R}^{d \times r}$. This parameter $r$ allows one to set the dimensionality of the latent representation of the RAE, if $r = d$ then the RAE reduces to $\mathrm{RAE}(\boldsymbol{x}) = \exp_{\boldsymbol{z}}^{\varphi}\left(\log_{\boldsymbol{z}}^{\varphi}(\boldsymbol{x})\right)$.

To learn a RAE, one needs to first construct a diffeomorphism and define an objective function. In Diepeveen (2024) diffeomorphisms are constructed by,

$$\varphi := [\psi^{-1}, \boldsymbol{I}_{d-d'}] \circ \phi \circ \boldsymbol{O} \circ T_{\boldsymbol{z}}, \tag{35}$$

where $\psi : U \to \mathbb{R}^{d'}$ is a chart on a (geodesically convex) subset $U \subset \mathcal{M}^{d'}$ of a $d'$-dimensional Riemannian manifold $(\mathcal{M}^{d'}, (\cdot, \cdot)_{\mathcal{M}_d'})$, $\phi : \mathbb{R}^d \to \mathbb{R}^d$ is a real-valued diffeomorphism, $\boldsymbol{O} \in \mathbb{O}(d)$ is an orthogonal matrix, and $T_{\boldsymbol{z}} : \mathbb{R}^d \to \mathbb{R}^d$ is given by $T_{\boldsymbol{z}}(\boldsymbol{x}) = \boldsymbol{x} - \boldsymbol{z}$. The learnable diffeomorphism $\varphi := \varphi_{\boldsymbol{\theta}}$ is constructed through parameterizing $\phi := \phi_{\boldsymbol{\theta}}$ by an invertible residual network Behrmann et al. (2019).

## A.3 LEARNING ISOMETRIES WITH RIEMANNIAN AUTO-ENCODERS

After constructing the diffeomorphism and Riemannian Auto-Encoder, one can learn an isometry by find the parameters $\boldsymbol{\theta}$ of $\varphi_{\boldsymbol{\theta}}$ in Diepeveen (2024) through minimizing the objective,

$$\mathcal{L}(\boldsymbol{\theta}) = \frac{1}{N(N-1)} \sum_{i,j=1, i \neq j}^{N(N-1)} \left(d_{\mathbb{R}^d}^{\varphi_{\boldsymbol{\theta}}}(\boldsymbol{x}_i, \boldsymbol{x}_j) - d_{i,j}\right)^2 \qquad \text{(\textbf{global isometry loss})}$$

$$+ \alpha_{\mathrm{sub}} \frac{1}{N} \sum_{i=1}^{N} \left\| \begin{bmatrix} \boldsymbol{I}_{d-d'} & \emptyset \\ \emptyset & \boldsymbol{0}_{d'} \end{bmatrix} (\phi_{\boldsymbol{\theta}} \circ \boldsymbol{O} \circ T_{\boldsymbol{z}})(\boldsymbol{x}_i) \right\|_1 \qquad \text{(\textbf{submanifold loss})}$$

$$+ \alpha_{\mathrm{iso}} \frac{1}{N} \sum_{i=1}^{N} \left\| \left(\left(\boldsymbol{e}^j, \boldsymbol{e}^{j'}\right)_{\boldsymbol{x}_i}^{\varphi_{\boldsymbol{\theta}}}\right)_{j,j'=1}^{d} - \boldsymbol{I}_d \right\|_F^2, \qquad \text{(\textbf{local isometry loss})}$$

where $\| \cdot \|_F$ is the Frobenius norm and $\left(\left(\boldsymbol{e}^j, \boldsymbol{e}^{j'}\right)_{\boldsymbol{x}_i}^{\varphi_{\boldsymbol{\theta}}}\right)_{j,j'=1}^{d}$ denotes a $d$-dimensional matrix just as $(\boldsymbol{A}_{ij})_{i,j=1}^{d}$ denotes a matrix.

First, the **global isometry loss** takes global geometry into account, ensuring that the learned distances under the diffeomorphism $\varphi_{\boldsymbol{\theta}}$ approximate the true pairwise distances $d_{i,j}$ between data points. Second, the **submanifold loss** enforces that the data manifold is mapped to $\mathcal{M} = \mathcal{M}^{d'} \times \mathbb{R}^{d-d'}$, preserving the submanifold structure of the data in the latent space. Finally, the **local isometry loss** enforces local isometry, ensuring that small-scale distances and local geometry are preserved under the transformation, which is critical for maintaining the intrinsic geometric properties of the data during dimensionality reduction. For further details on the implementation and theoretical considerations, see Diepeveen (2024).

### A.4 CONDITIONAL FLOW MATCHING

To achieve the goal of accurate generative modeling on data manifolds through isometric learning, we first need to understand generative modeling on Euclidean spaces. We do this through summarizing CFM Lipman et al. (2022), a commonly used and effective framework for learning CNFs for generative modeling for Euclidean data Chen et al. (2018). CFM is a method designed to map a simple base distribution to a target data distribution by learning a time-dependent vector field. The fundamental goal of Flow Matching (FM) is to align a target probability path $p_t(\boldsymbol{x})$ with a vector field $u_t(\boldsymbol{x})$, which generates the desired distribution. The FM objective is defined as follows:

$$\mathcal{L}_{\text{FM}}(\boldsymbol{\eta}) = \mathbb{E}_{t,p_t(\boldsymbol{x})}\|v_t(\boldsymbol{x};\boldsymbol{\eta}) - u_t(\boldsymbol{x})\|^2, \tag{36}$$

where $\boldsymbol{\eta}$ represents the learnable parameters of the neural network that parameterizes the vector field $v_t(\boldsymbol{x};\boldsymbol{\eta})$, and $t \sim \mathcal{U}(0,1)$ is uniformly sampled. However, a significant challenge in FM is the intractability of constructing the exact path $p_t(\boldsymbol{x})$ and the corresponding vector field $u_t(\boldsymbol{x})$.

To address this Lipman et al. (2022) introduce CFM, a more practical approach by constructing the probability path and vector fields in a conditional manner. The CFM objective is then formulated by marginalizing over the data distribution $q(\boldsymbol{x}_1)$ and considering the conditional probability paths:

$$\mathcal{L}_{\text{CFM}}(\boldsymbol{\eta}) = \mathbb{E}_{t,q(\boldsymbol{x}_1),p_t(\boldsymbol{x}|\boldsymbol{x}_1)}\|v_t(\boldsymbol{x};\boldsymbol{\eta}) - u_t(\boldsymbol{x}|\boldsymbol{x}_1)\|^2. \tag{37}$$

A key result, as established in Theorem 2 of Chen & Lipman (2024), is that the gradients of the CFM objective with respect to the parameters $\boldsymbol{\eta}$ are identical to those of the original FM objective, i.e.,

$$\nabla_{\boldsymbol{\eta}}\mathcal{L}_{\text{FM}}(\boldsymbol{\eta}) = \nabla_{\boldsymbol{\eta}}\mathcal{L}_{\text{CFM}}(\boldsymbol{\eta}), \tag{38}$$

ensuring that optimizing the CFM objective yields the same result as the original FM objective. This enables effective train of the neural network without needing direct access to the intractable marginal probability paths or vector fields.

Given a sample $\boldsymbol{x}_1$ from the data distribution $q(\boldsymbol{x}_1)$, we define a conditional probability path $p_t(\boldsymbol{x}|\boldsymbol{x}_1)$ [3]. This path starts at $t = 0$ from a simple distribution, typically a standard Gaussian, and approaches a distribution concentrated around $\boldsymbol{x}_1$ as $t \to 1$:

$$p_t(\boldsymbol{x}|\boldsymbol{x}_1) = \mathcal{N}(\boldsymbol{x}|\mu_t(\boldsymbol{x}_1), \sigma_t(\boldsymbol{x}_1)^2\mathbf{I}), \tag{39}$$

where $\mu_t(\boldsymbol{x}_1) : [0,1] \times \mathbb{R}^d \to \mathbb{R}^d$ is the time-dependent mean, and we denote the time-dependent standard deviation as $\sigma_t(\boldsymbol{x}_1) : [0,1] \times \mathbb{R} \to \mathbb{R}_{>0}$. For simplicity, we set $\mu_0(\boldsymbol{x}_1) = \mathbf{0}$ and $\sigma_0(\boldsymbol{x}_1) = 1$, ensuring that all conditional paths start from the same standard Gaussian distribution. At $t = 1$, the path converges to a distribution centered at $\boldsymbol{x}_1$ with a small standard deviation $\sigma_{\min}$.

The corresponding conditional vector field $u_t(\boldsymbol{x}|\boldsymbol{x}_1)$ can be defined by considering the flow:

$$\chi_t(\boldsymbol{x}) = \sigma_t(\boldsymbol{x}_1)\boldsymbol{x} + \mu_t(\boldsymbol{x}_1), \tag{40}$$

which maps a sample from the standard Gaussian to a sample from $p_t(\boldsymbol{x}|\boldsymbol{x}_1)$. The vector field $u_t(\boldsymbol{x}|\boldsymbol{x}_1)$ that generates this flow, as proven by Lipman et al. (2022) in Theorem 3, is given by:

$$u_t(\boldsymbol{x}|\boldsymbol{x}_1) = \frac{\sigma_t'(\boldsymbol{x}_1)}{\sigma_t(\boldsymbol{x}_1)}\left(\boldsymbol{x} - \mu_t(\boldsymbol{x}_1)\right) + \mu_t'(\boldsymbol{x}_1), \tag{41}$$

where the primes denote derivatives with respect to time to stay consistent with the original papers notation.

In this work, we choose to use the optimal transport (OT) formulation of CFM. Here, the mean $\mu_t(\boldsymbol{x}_1)$ and standard deviation $\sigma_t(\boldsymbol{x}_1)$ are designed to change linearly in time, offering a straightforward interpolation between the base distribution and the target distribution. Specifically, the mean

---

[3]In this work, we use two types of indexing: $x_t$ to denote time indices and $x_i$ for different data points. It should be clear from the context which indexing is being used.

and standard deviation are defined as:

$$\mu_t(\boldsymbol{x}_1) = t\boldsymbol{x}_1, \quad \sigma_t(\boldsymbol{x}_1) = 1 - (1 - \sigma_{\min})t. \tag{42}$$

This linear path results in a vector field $u_t(\boldsymbol{x}|\boldsymbol{x}_1)$ given by:

$$u_t(\boldsymbol{x}|\boldsymbol{x}_1) = \frac{\boldsymbol{x}_1 - (1 - \sigma_{\min})\boldsymbol{x}}{1 - (1 - \sigma_{\min})t}. \tag{43}$$

The corresponding conditional flow that generates this vector field is:

$$\chi_t(\boldsymbol{x}) = (1 - (1 - \sigma_{\min})t)\boldsymbol{x} + t\boldsymbol{x}_1. \tag{44}$$

This OT path is optimal in the sense that it represents the displacement map between the two Gaussian distributions $p_0(\boldsymbol{x}|\boldsymbol{x}_1)$ and $p_1(\boldsymbol{x}|\boldsymbol{x}_1)$ Lipman et al. (2022).

The final CFM loss under this OT formulation is derived by substituting the above vector field and flow into the general CFM objective (Equation 37) and reparameterizing $p_t(\boldsymbol{x}|\boldsymbol{x}_1)$ in terms of $\boldsymbol{x}_0$. This yields the following objective function:

$$\mathcal{L}_{\text{CFM}}(\boldsymbol{\eta}) = \mathbb{E}_{t,q(\boldsymbol{x}_1),p(\boldsymbol{x}_0)} \left\| v_t(\chi_t(\boldsymbol{x}_0); \boldsymbol{\eta}) - \frac{\boldsymbol{x}_1 - (1 - \sigma_{\min})\boldsymbol{x}_0}{1 - (1 - \sigma_{\min})t} \right\|^2. \tag{45}$$

This formulation is advantageous because the OT paths ensure that particles move in straight lines and with constant speed, leading to simpler and more efficient regression tasks compared to traditional diffusion-based methods. We use the OT-CFM objective in this work when we refer to CFM.

## A.5 RIEMANNIAN FLOW MATCHING

The next step toward generation on data manifolds is understanding generation on manifolds with closed form geometric mappings. RFM aims to do exactly this by generalizing CFM to Riemannian manifolds Chen & Lipman (2024). Assume a complete, connected and smooth manifold $\mathcal{M}$ endowed with a Riemannian metric $(\cdot, \cdot)^{\mathcal{M}}$. We are given a set of training samples $\boldsymbol{x}_1 \in \mathcal{M}$ from some unknown data distribution $q(\boldsymbol{x}_1)$ on the manifold. Then the goal is to learn a parametric map $\rho : \mathcal{M} \to \mathcal{M}$ that pushes a simple base distribution $p$ to the data distribution $q$. To achieve RFM Chen & Lipman (2024) reparameterize the conditional flow as

$$\boldsymbol{x}_t = \chi_t(\boldsymbol{x}_0 | \boldsymbol{x}_1), \tag{46}$$

where $\chi_t(\boldsymbol{x}_0 | \boldsymbol{x}_1)$ is the solution to the ordinary differential equation (ODE) defined by a time-dependent conditional vector field $u_t(\boldsymbol{x} | \boldsymbol{x}_1) \in \mathcal{T}_{\boldsymbol{x}} \mathcal{M}$ that is tangent to the manifold $\mathcal{M}$. The initial condition is set as $\chi_0(\boldsymbol{x}_0 | \boldsymbol{x}_1) = \boldsymbol{x}_0$.

This formulation leads to the RFM objective, which ensures that the vector field $u_t(\boldsymbol{x} | \boldsymbol{x}_1)$ learned by the model lies entirely within the tangent space of the manifold at each point $\boldsymbol{x}_t \in \mathcal{M}$:

$$\mathcal{L}_{\text{RFM}}(\boldsymbol{\eta}) = \mathbb{E}_{t, q(\boldsymbol{x}_1), p(\boldsymbol{x}_0)} \Big( \| v_t(\boldsymbol{x}_t; \boldsymbol{\eta}) - u_t(\boldsymbol{x}_t | \boldsymbol{x}_1) \|_{\mathbf{x}_t}^{\mathcal{M}} \Big)^2, \tag{47}$$

where $\| \cdot \|_{\mathcal{M}}$ is the norm enduced by the Riemannian metric $(\cdot, \cdot)^{\mathcal{M}}$. For manifolds with closed-form geodesic expressions, a simulation-free objective can be formulated using exponential and logarithmic maps. This approach allows models to be trained without numerically simulating particle trajectories, leveraging closed-form geodesics and mappings to directly compute vector fields and transport paths. In this case, $\boldsymbol{x}_t$ can be defined by the geodesic between $\boldsymbol{x}_1$ and $\boldsymbol{x}_0$ and can be explicitly expressed as

$$\boldsymbol{x}_t = \gamma_{\boldsymbol{x}_1, \boldsymbol{x}_0}^{\mathcal{M}} \big( \kappa(t) \big) = \exp_{\boldsymbol{x}_1}^{\mathcal{M}} \big( \kappa(t) \log_{\boldsymbol{x}_1}^{\mathcal{M}} (\boldsymbol{x_0}) \big), \tag{48}$$

with monotonically decreasing differentiable function $\kappa(t)$ satisfying $\kappa(0) = 1$ and $\kappa(1) = 0$ acting as a scheduler. Furthermore, the tangent vector field $u_t(\boldsymbol{x} | \boldsymbol{x}_1)$ can be evaluated through,

$$u_t(\boldsymbol{x} | \boldsymbol{x}_1) = \dot{\gamma}_{\boldsymbol{x}_1, \boldsymbol{x}_0}^{\mathcal{M}} \big( \kappa(t) \big) = \frac{d}{dt} \exp_{\boldsymbol{x}_1}^{\mathcal{M}} \big( \kappa(t) \log_{\boldsymbol{x}_1}^{\mathcal{M}} (\boldsymbol{x_0}) \big) \tag{49}$$

The objective function is then given by:

$$\mathcal{L}_{\text{RFM}}(\boldsymbol{\eta}) = \mathbb{E}_{t, q(\boldsymbol{x}_1), p(\boldsymbol{x}_0)} \Big( \big\| v_t(\gamma_{\boldsymbol{x}_0, \boldsymbol{x}_1}^{\mathcal{M}} \big( \kappa(t) \big); \boldsymbol{\eta}) - \dot{\gamma}_{\boldsymbol{x}_0, \boldsymbol{x}_1}^{\mathcal{M}} \big( \kappa(t) \big) \big\|_{\mathbf{P}}^{\mathcal{M}} \Big)^2 \tag{50}$$

Constructing a simulation-free objective for RFM on general geometries presents significant challenges due to the absence of closed-form expressions for essential geometric operations, such as exponential and logarithmic maps, or geodesics. These operations are crucial for defining and efficiently evaluating the objective but are often computationally intensive to approximate without closed-form solutions. For a list of examples of manifolds with closed-form geometric mappings, see the appendix of Chen & Lipman (2024).

In the absence of such closed-form solutions, existing methods tackle these difficulties by either learning a metric that constrains the generative trajectory to align with the data support Kapusniak et al. (2024) or by assuming a metric with easily computable geodesics on the data manifold Chen & Lipman (2024). However, learning a metric can be problematic as it may lead to overfitting or fail to capture the true geometry of the data, particularly when the data manifold is complex or poorly understood. On the other hand, assuming a simple metric with computable geodesics can oversimplify the problem, resulting in models that inadequately represent the underlying data structure. To overcome these challenges, we introduce Pullback Flow Matching in section 3.

## B   NEURAL ODEs PARAMETERIZE DIFFEOMORPHISMS

We can verify that this defines a diffeomorphism by using Theorem C.15 of Younes (2010). According to Theorem C.15, for $\phi_\theta$ to be a diffeomorphism, the vector field $f$ must satisfy $f \in L^1([0,1], C^1_{(0)}(\Omega, \mathbb{B}))$, where $\Omega$ is the domain of the vector field and $\mathbb{B}$ is a Banach space representing the target space.

In our case, $f$ is composed of smooth and continuously differentiable functions due to the MLP parameterization, ensuring $f$ is also smooth and continuously differentiable. Additionally, we enforce local isometry by regularizing the Jacobian of $f_\theta$, which guarantees local regularity of $f$ in the data domain (see **stability regularization**). Thus, $f$ meets the required conditions and $\phi_\theta$ defines a proper diffeomorphism.

## C   MANIFOLD AND METRIC SELECTION

Isometric learning requires three key choices to be made, first one needs to choose the Riemannian metric of the data manifold $(\cdot, \cdot)^\mathcal{D}$, second one needs to choose both the latent (sub)manifold and its Riemannian metric $(\mathcal{M}_{d'}, (\cdot, \cdot)^{\mathcal{M}_{d'}})$ and finally one needs to choose the dimensionality $d'$. Technically one also needs to assume a metric on $\mathbb{R}^{d-d'}$, but in this work we assume a Euclidean metric $(\cdot, \cdot)_2$ throughout all our experiments.

There are several options when selecting the metric on the data manifold $(\cdot, \cdot)^\mathcal{D}$. One can choose for example a locally euclidean approximation through Isomap Tenenbaum et al. (2000) or a more noise-robust geodesic approximation Little et al. (2022). One can also design a metric to create a latent space [4] structured based on properties of the data one cares about, we show how in subsection 5.4. In this work, we focus on using a proper metric and defer the exploration of learning with pseudo-metrics to future research.

When selecting a latent Riemannian (sub)manifold and metric it is crucial to select $\mathcal{M}_{d'}$ such that $\mathcal{M} = \mathcal{M}_{d'} \times \mathbb{R}^{d-d'}$ it is diffeomorphic to the data manifold $\mathcal{D}$. This ensures that the latent space of the RAE can effectively capture the intrinsic structure of the data. The manifold should be chosen based on its ability to accommodate the data's periodicity, curvature, and dimensionality. This alignment is essential for accurately representing the data manifold within the latent space. Unless otherwise stated we assume $\mathcal{M}_{d'} = \mathbb{R}^{d'}$. Additionally, one should select the Riemannian geometry of $(\mathcal{M}_{d'}, (\cdot, \cdot)^{\mathcal{M}_{d'}})$ such that geometric mappings can be explicitly defined in closed form. A list of manifolds with closed form geometric mappings can be found in the appendix of Chen & Lipman (2024). Unless otherwise states we select $(\cdot, \cdot)^{\mathcal{M}_{d'}} = (\cdot, \cdot)_2$.

Finally, $d'$, the dimensionality of the latent space, is a hyperparameter that could be tuned through iterative testing. Techniques such as Isomap Tenenbaum et al. (2000) or equivalents on other manifolds such as hyperbolic space Cvetkovski & Crovella (2011) can be employed to evaluate various dimensional and Riemannian geometric settings and determine the optimal $d'$ that balances model complexity with the ability to accurately capture the data manifold's structure.

---

[4]In this text we refer to the latent space as the concept in machine and representation learning, technically its a latent manifold endowed with a Riemannian metric, not a vector space.

# D  DATA DESCRIPTION

In this work we use several datasets, synthetic, simulated and experimental. Here we describe them in order of appearance in the experiments.

## D.1  ARCH DATASET

We create a dataset in the spirit of Tong et al. (2020). We sample $n = 500$ data points uniform on the line $[-1, 1]$ ($x_i \sim \mathcal{U}(-1, 1)$), wrap this line around the unit half circle and add normally distributed noise with $\sigma = 0.1$, i.e.

$$y_{i,1} = \sin(0.5\pi x_i) + a_{i,1}, \quad y_{i,2} = \cos(0.5\pi x_i) + a_{i,2} \text{ for } a_{i,j} \sim \mathcal{N}(0, 0.1^2). \tag{51}$$

An example of the dataset can be found in Figure 5.

## D.2  ADENYLATE KINASE (AK)

We consider the time-normalized open-to-close transition of AK. This is a dataset from coarse-grained molecular dynamics simulations consisting of $n = 102$ conformations of 214 amino-acids in $3D$, samples of the trajectory can be found in Figure 8.

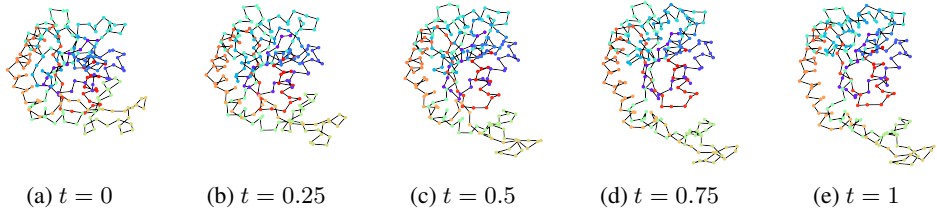

(a) $t = 0$   (b) $t = 0.25$   (c) $t = 0.5$   (d) $t = 0.75$   (e) $t = 1$

Figure 8: Example of the open-to-close transition of adynalate kinase protein.

## D.3  INTESTINAL FATTY ACID BINDING PROTEIN

The second protein dynamics dataset is that of $n = 500$ conformations of I-FABP in water. The datasets comes from simulations in CHARMM of 500 picoseconds (ps) with a 2 femtoseconds (fs) timestep. The data can be found on mdanalysis.org.

## D.4  GRAMPA DATASET

The giant repository of AMP activities (GRAMPA) dataset Witten & Witten (2019) is a compilation of peptides and their antimicrobial activity against various bacteria, including *E. coli* and *P. aeruginosa*. It includes data on peptide sequences, target bacteria, bacterial strains, and minimal inhibitory concentration (MIC) values, with additional columns providing details on sequence modifications and data sources. The dataset was created to support deep learning models aimed at predicting the antimicrobial effectiveness of peptides. The dataset is available here. In our experiments we follow the preprocessing pipeline from Szymczak et al. (2023) and use only the sequence data and the corresponding MIC scores. After preprocessing we are left with $n = 3444$ sequences of maximum 25 amino-acids with tested antimicrobial activity against E. coli.

# E  TRAINING PROCEDURE

We explain the training procedure and hyperparameter settings for each of the experiments in section 5 in further detail for reproducibility. In all experiments the datasets where split into train and test sets. We apply early stopping and present the model with the lowest average loss on the test data.

## E.1  ABLATION STUDY

For details of hyperparameter settings for the ablation study see Table 5.

Table 5: Hyperparameter settings for ablation study of RAE on the ARCH, AK and I-FABP datasets.

| Hyperparameter | ARCH | AK | I-FABP |
|---|---|---|---|
| Epochs | 1000 | 1000 | 1000 |
| Learning Rate | 0.0001 | 0.0001 | 0.0001 |
| Optimizer | Adam | Adam | Adam |
| Train/Test Split | 0.8/0.2 | 0.8/0.2 | 0.8/0.2 |
| $n_{steps}$ | 10 | 10 | 10 |
| Seed | 0 | 0 | 0 |
| Number of Layers | 5 | 5 | 5 |
| $\alpha_1$ | 1.0 | 1.0 | 1.0 |
| $\alpha_2$ | $[0.0, 1.0]$ | $[0.0, 5.0]$ | $[0.0, 5.0]$ |
| $\alpha_3$ | 1.0 | 1.0 | 1.0 |
| $\alpha_4$ | $[0.0, 0.01]$ | $[0.0, 0.005]$ | $[0.0, 0.1]$ |
| $d'$ | 1 | 1 | 1 |
| Hidden Units | 64 | $214 \cdot 3 + 1$ | $131 \cdot 3 + 1$ |
| Number of Neighbors | 5 | 2 | 4 |
| Batch Size | 64 | 16 | 64 |
| Warmup | 50 | 400 | 200 |

Specific hyperparameters worth mentioning are $n_{steps}$ which is the number of Runge-Kutta steps we use in our Neural ODE, *Number of Layers* is the number of layers of the MLP with swish activation function for the vector field of the Neural ODE. The *Number of Neighbors* is the hyperparameter used to calculate the shortest paths over the nearest neighbors graph for the Isomap geodesics in `sklearn` and the *Warmup* is the number of epochs we train with $\alpha_1, \alpha_2 = 0$ to first learn a lower dimensional representation.

## E.2 INTERPOLATION EXPERIMENTS

For details of hyperparameter settings for the interpolation experiments of $(\cdot, \cdot)^{\mathcal{M}}$- and $(\cdot, \cdot)^{\mathcal{M}_{d'}}$-interpolation see Table 6 and for the $(\beta$-)VAEs see Table 7.

Table 6: Hyperparameter settings for interpolation experiments for $(\cdot, \cdot)^{\mathcal{M}}$- and $(\cdot, \cdot)^{\mathcal{M}_{d'}}$-interpolation on the ARCH, AK and I-FABP datasets.

| Hyperparameters | ARCH | AK | I-FABP | |
|---|---|---|---|---|
| Epochs | 1000 | 1000 | 1000 | |
| Learning Rate | 0.0001 | 0.0001 | 0.0001 | |
| Optimizer | Adam | Adam | Adam | |
| Train/Test Split | 0.8/0.2 | 0.8/0.2 | 0.8/0.2 | |
| $n_{steps}$ | 10 | 10 | 10 | |
| Seed | 0 | 0 | 0 | |
| Number of Layers | 5 | 5 | 5 | |
| $\alpha_1$ | 1.0 | 1.0 | 1.0 | |
| $\alpha_2$ | 5.0 | 5.0 | 5.0 | |
| $\alpha_3$ | 1.0 | 1.0 | 1.0 | |
| $\alpha_4$ | 0.001 | 0.005 | 0.1 | |
| $d'$ | 1 | 1 | 1 | 5 |
| Hidden Units | 64 | $214 \cdot 3 + 1$ | $131 \cdot 3 + 1$ | |
| Number of Neighbors | 5 | 2 | 4 | |
| Batch Size | 64 | 16 | 64 | |
| Warmup | 50 | 400 | 200 | |
| $n_{parameters}$ | 17282 | 2486480 | 934961 | |

Table 7: Hyperparameter settings for interpolation experiments for $(\beta$-)VAE on the ARCH dataset. VAEs have $\beta = 1.0$, $\beta$-VAEs have $\beta = 10.0$.

| Hyperparameters | ARCH | AK | I-FABP |
|---|---|---|---|
| Epochs | 1000 | 1000 | 1000 |
| Learning Rate | 0.0001 | 0.0001 | 0.0001 |
| Optimizer | Adam | Adam | Adam |
| Train/Test Split | 0.8/0.2 | 0.8/0.2 | 0.8/0.2 |
| Seed | 0 | 0 | 0 |
| Number of Encoder Layers | 5 | 5 | 5 |
| Number of Decoder Layers | 5 | 5 | 5 |
| Hidden Units | 64 | $214 \cdot 3 + 1$ | $131 \cdot 3 + 1$ |
| Beta | $[1.0, 10.0]$ | $[1.0, 10.0]$ | $[1.0, 10.0]$ |
| $d'$ | 1 | 1 | 1 |
| Batch Size | 64 | 16 | 64 |
| $n_{parameters}$ | 34184 | 4555655 | 1712324 |

### E.3 GENERATION EXPERIMENTS

Table 8: Hyperparameter settings for CFM, PFM and $d'$-PFM for generation experiments. The same isometry $\varphi_{\boldsymbol{\theta}}$ of the interpolation experiments is used for the PFM and $d'$-PFM.

| Hyperparameter | ARCH | | | I-FABP | | |
|---|---|---|---|---|---|---|
| | CFM | PFM | $d'$-PFM | CFM | PFM | $d'$-PFM |
| Epochs | 5000 | 5000 | 5000 | 5000 | 5000 | 5000 |
| Learning Rate | 0.0005 | 0.0005 | 0.0005 | 0.001 | 0.001 | 0.0005 |
| Scheduler | Cosine | Cosine | Cosine | Cosine | Cosine | Cosine |
| Minimum Learning Rate | $5.0 \cdot 10^{-6}$ | $5.0 \cdot 10^{-6}$ | $5.0 \cdot 10^{-6}$ | $1.0 \cdot 10^{-5}$ | $1.0 \cdot 10^{-5}$ | $5.0 \cdot 10^{-6}$ |
| Train/Test Split | 0.8/0.2 | 0.8/0.2 | 0.8/0.2 | 0.8/0.2 | 0.8/0.2 | 0.8/0.2 |
| Seed | 0 | 0 | 0 | 0 | 0 | 0 |
| Number of Layers | 10 | 10 | 10 | 10 | 10 | 10 |
| Hidden Units | 64 | 64 | 16 | $131 \cdot 3 + 1$ | $131 \cdot 3 + 1$ | $131 \cdot 3 + 1$ |
| Batch Size | 64 | 64 | 64 | 64 | 64 | 64 |
| $n_{\text{simulation steps}}$ | 10 | 10 | 10 | 10 | 10 | 10 |

### E.4 DESIGNABLE LATENT MANIFOLDS FOR NOVEL PROTEIN ENGINEERING

In Table 9 one can find the settings for training the isometry on the GRAMPA dataset for the protein sequence design experiments. Specific hyperparameter worth mentioning is the embedding dimensions, we use an embedding layer from the `Flax` library to embed the discrete sequences into a continuous space and use a sign-cosine positional embedding, to embed the location in the sequence of the amino acids in the data.

Table 9: Hyperparameter settings for protein design experiments of the RAEs on the GRAMPA dataset.

| Hyperparameters | Setting |
|---|---|
| Epochs | 1000 |
| Learning Rate | 0.0001 |
| Optimizer | Adam |
| Train/Test Split | 0.8/0.2 |
| $n_{steps}$ | 10 |
| Seed | 0 |
| Number of Layers | 5 |
| Embedding dimension | 8 |
| $\alpha_1$ | 5.0 |
| $\alpha_2$ | 5.0 |
| $\alpha_3$ | 5.0 |
| $\alpha_4$ | 0.05 |
| $d'$ | 128 |
| Hidden Units | 512 |
| Batch Size | 128 |
| Warmup | 100 |

