# OpenReview forum: "Pullback Flow Matching on Data Manifolds"
_ICLR.cc/2025/Conference — Submitted to ICLR 2025_

### Official Review · Reviewer_7H7T · 2024-10-29

**Soundness:** 2
**Presentation:** 2
**Contribution:** 2
**Rating:** 5
**Confidence:** 4

**Summary:**

This paper proposes Pullback Flow Matching, a novel framework for generative modeling on data
with manifold structures. Building on the previous study, this work leverages pullback geometry
to define a new metric on the entire ambient space $\mathbb{R}^d$, by learning an isometry that preserves the
geometric structure of the data manifold $\mathcal{D}$ on the latent manifold $\mathcal{M}$. The corresponding metric of
the assumed latent manifold $\mathcal{M}$ is then used to perform Riemannian Flow Matching (RFM) on the
latent manifold. The effectiveness of this method is demonstrated through experiments on synthetic
data, molecular dynamics data, and experimental peptide sequences.

Main contributions:
1. A novel generative modeling framework that combines isometric learning and Flow Matching.
2. An objective for isometric learning that only relies on distance measure on data manifold.
3. Improved parameterization of isometries via neural ODEs.

**Strengths:**

###  Originality
This paper proposes a new generative modeling approach for datasets with manifold structure. The approach combines flow matching and isometric learning, which preserve the intrinsic geometric structure of the data manifold in the latent space.

### Quality

The research question studied in this paper, i.e. generative modeling for data with manifold structure, is definitely interesting. The proposed learning approach, i.e. isometric learning and flow matching, as well as the training objectives, are reasonable. The experiments are discussed with sufficient details.

### Clarity

Overall, the presentation of the research question, the learning approach and the numerical experiments of this paper is  clear.

### Significance

The proposed approach may have certain advantages over existing methods for dataset with manifold structure.

**Weaknesses:**

1. In the numerical experiments (Section 5), a relatively large part of the text focuses on isometry learning, and less effort is made on the study of the generation part. Results in Table 3 of Section 5.3 show that 1-PFM is better than CFM/PFM, but 1-PFM is solving a modified/simplified generative task instead of generating true data distribution. The same is true for the analogue generation in Section 5.4. Given these facts, the referee is afraid that the presented material may not be sufficient to firmly support some of the statements/claims made in the the abstract and introduction.

1. Section 2 and 3 consider a general setting, but the practical implementation only addresses a special case, which may cause confusion. Concretely, while the latent space is defined as a general manifold $\mathcal{M}$ in Section 2 and 3, it is chosen as the Euclidean space $\mathbb{R}^d$ in practical computations. In particular, if I understand correctly, Eq. (6) and (7) are actually flow matching losses in the latent space $\mathbb{R}^d$ or $\mathbb{R}^{d'}$ with the standard Euclidean metric. Writing them in a general form does not help readers understand the idea.

**Questions:**

1. What is the dimension $d$ of the data in GRAMPA dataset in Section 5.4? Why is $d'$ chosen as 128? This should be explained.

1. In Eq. (8), how do you choose $\mu$ in $T_{\mu}$? What role does this translation play? How is $\psi$ in Eq. (8) selected, and is it simply the identity mapping when $\mathcal{M}'$ is Euclidean space $\mathbb{R}^{d'}$? Could you provide more related details in the paper?

1. In the submanifold loss (line 279), why is the notation $T_{\mu}$ changed to $T_{z}$? Is $z$ related to $z_i$ in the stability regularization (line 282)?

1. It seems that choosing the latent manifold $\mathcal{M}$ as $\mathbb{R}^d$ with Euclidean metric is  sufficient (to approximate the distance measure $(\cdot,\cdot)^{\mathcal{D}}$ on the data manifold using an isometric mapping). In fact, this is how it was selected in the experiments. Why not present equations in Section 3 in a more specific form? The presentation there in general form makes it harder to understand.

1. In all the experiments except the last one, $d'$ is chosen to be 1. Is this choice made because the datasets are essentially concentrated on one-dimensional manifolds? How to choose $d'$ in more general cases? Can the author consider examples where $d'$ is greater than 1?

1. The sizes of the datasets in the experiments are quite small, sometimes even lower than the dimension of the data. Can the method apply to large datasets? Training the isometries (using the global isometry loss and graph matching loss) requires iterating over all point pairs, which seems to be computationally intensive when the batch size is large. Can the authors discuss the computational cost of isometric learning, especially when the size of the dataset is large?

 1.   There are several typos throughout the paper.

- Line 070, the sentence doesn't read correctly.
- Figure 2, the direction of the arrow connecting $x_1$ and $\varphi(x_1)$ seems to be wrong.
- Line 321:  "as well ass" should be "as well as".
- Line 284: "induces" should be "induced".
- Line 1043: Missing citation.
- Line 279: To align with Eq (8), $I_{d-d'}$ and $0_{d'}$ should be switched in the submanifold loss.
- The initial letter of the names in references, e.g. poincare, ode, riemannian, jacobian, bayes, should be capitalized.

---

> ### Author Response · Authors · 2024-11-24
> **[1/3]**
>
> Dear reviewer,
>
> Thank you for taking the time to read and review our paper. We appreciate your thoughtful feedback, which has helped us improve the explanation of the hyperparameter selection, the method and the presentation of our equations. Below, we provide detailed responses to your points of feedback and questions.
>
> Strengths
> We sincerely thank the reviewer for recognizing the originality, quality, clarity, and potential significance of our work. We appreciate the reviewer’s recognition of our combination of flow matching and isometric learning as a meaningful contribution to generative modeling for datasets with manifold structures.
>
> Weakness 1: Focus more on generative modeling part of the paper to firmly support the claims in the abstract and introduction.
> Thank you for raising this important concern. We value scientific integrity and are committed to ensuring that all claims are well-supported by evidence. We have been careful in acknowledging prior works and formulating our claims. In this regard, could you please clarify which specific claims in the abstract and introduction are of concern?
>
> In the abstract and introduction, we primarily claim that our method:
> 1. Enables efficient generation and precise interpolation in latent space through pullback geometry and isometric learning
> 2. Facilitates closed-form mappings while preserving the manifold's geometry
> 3. Achieves improved manifold learning and generative performance through enhanced isometric learning
>
> The concern seems to be about generating on lower-dimensional manifolds, which we agree is indeed an easier task. This is precisely why our isometric learning approach is so fitting for the generative modeling context, as it separates out the manifold learning and density estimation aspects of generative modeling. The key comparison then is between generating the raw data directly versus doing this using the geometry of the learned data manifold (what we propose). We show that the latter is beneficial, and even more so when we constrain the generation to the projected low-dimensional manifold, as this indeed simplifies the generative component by removing noise directions.
>
> Weakness 2: General formulation in Sections 2 and 3 may confuse readers as practical implementation focuses on a Euclidean latent space.
> We value simplicity, correctness, and clarity in our scientific writing and equations. We understand the reviewer’s concern regarding the potential for confusion due to the general form of the equations in Sections 2 and 3, particularly since the practical implementation focuses on a Euclidean latent space $\mathbb{R}^d$. We appreciate your feedback and have made adjustments to clarify the rationale behind this choice.
>
> Our decision to formulate the equations in a general form stems from the flexibility and scope of our theoretical framework. By expressing the latent space as a general manifold $\mathcal{M}$, we aim to demonstrate how our approach can be applied to datasets with different underlying geometries, not limited to Euclidean spaces. This generality establishes a direct connection between our work and other manifold learning methods that address broader settings. For example, the use of the pullback flow matching framework is inherently agnostic to the choice of latent space geometry. This generality ensures that future extensions of our work to non-Euclidean latent spaces, such as hyperbolic or spherical geometries, do not require fundamentally new formulations. We believe this approach highlights the versatility of the framework and its potential applicability across diverse settings.
>
> That said, we recognize that presenting the equations in a general form may have caused some confusion regarding the practical implementation. To address this, we have revised the paper to clearly specify that our experiments are conducted with a latent manifold of $R^d$ and the Euclidean metric. We have explicitly noted this in the methods and notation sections, ensuring that the specific case used for practical computations is transparent to readers.
>
> We appreciate the reviewer’s insight on this point and are committed to balancing the clarity of our presentation with the general applicability of the theoretical framework.

---

> > ### Author Response · Authors · 2024-11-24
> > **[2/3]**
> >
> > Question 1: In GRAMPA dataset in Section 5.4, what is the dimension $d$ and $d'$ and why?
> > \textbf{Response to Question 1:}
> > We would like to thank the reviewer for noticing that the explanation regarding the dimensions $d$ and $d'$ in the GRAMPA dataset (Section 5.4) was lacking in the current version of the paper.
> >
> > The dataset involves protein sequences, where there are 20 amino acids and a special token representing the absence of an amino acid, resulting in 21 distinct amino acid states for each position in the sequence. With 25 positions, this creates a $25\times 21$-dimensional manifold. To work with this, we use an embedding layer with 8 dimensions, which projects the data into a $200$-dimensional latent space ($d = 200$).
> >
> > As for the selection of $d'$, we first run Isomap embedding to estimate the intrinsic dimensionality of the manifold. Specifically, to determine the optimal dimension $d'$, we test several different dimensions and compute the \textit{stress levels} for each embedding. Stress, in this case, refers to the difference between the pairwise distances in the high-dimensional space and those in the lower-dimensional embedding. Specifically, stress measures the discrepancy between the original pairwise distances $d_{ij}$ in the high-dimensional space and the corresponding distances $\hat{d}_{ij}$ in the embedded lower-dimensional space. Based on this estimate, we conduct a small grid search over different hyperparameter settings to fine-tune $d'$ and optimize the performance. This process resulted in selecting $d' = 128$ as the optimal choice for the GRAMPA dataset.
> >
> > To provide more context, the choice of $d'$ is crucial because the method aims to find a mapping that closely approximates the embedding produced by techniques like Isomap. The optimal dimensionality corresponds to the one where the stress (or the error in representing the data manifold) is minimized, as noted in Sonthalia et al., \textit{How Can Classical Multidimensional Scaling Go Wrong?} (2021). Using dimensions smaller or larger than the optimal value of $d'$ results in poorer performance, as it fails to preserve the geometric structure of the manifold properly. For datasets with known theoretical dimensions, such as the ARCH and Swiss roll datasets, $d'$ is selected based on the dataset's design. However, for experimental datasets like the protein dynamics trajectories, where the intrinsic dimension is not immediately known, we use the Isomap-based approach to estimate $d'$ and confirm it with a hyperparameter search.
> >
> > We have included a discussion of the procedure for selecting $d'$ in Appendix C of the manuscript for further clarity.
> >
> > Question 2: Choice of $\boldsymbol{\mu}$ and $\psi$ in Eq. (8)
> > In Eq. (8), the translation $\boldsymbol{\mu}$ is chosen to be a reference point on the manifold, specifically for deep learning purposes we choose the average of the set of points. This choice helps to position the manifold in a way that improves its alignment during the learning process. By translating the manifold to the centroid, we ensure that the learned representations are centered around a natural reference point, which aids in maintaining an interpretable latent space.
> >
> > The function $\psi$ in Eq. (8) is a chart, which serves as a local coordinate system mapping points from the manifold to Euclidean space. This chart $\psi$ allows for easier manipulation of the manifold’s geometry during optimization. When $\mathcal{M}'$ is Euclidean space (i.e., $\mathbb{R}^{d'}$), $\psi$ can be the identity mapping, as no additional transformation is required in Euclidean space. However, for more complex data types or non-Euclidean manifolds, $\psi$ can be a more complex transformation that captures the manifold’s geometry. We will provide additional details in the paper to clarify the roles of $\boldsymbol{\mu}$ and $\psi$.
> >
> > Question 3: Notation change from $T_{\boldsymbol{\mu}}$ to $T_{\boldsymbol{z}}$
> > The change in notation from $T_{\boldsymbol{\mu}}$ to $T_{\boldsymbol{z}}$ in the submanifold loss (line 279) was a typographical error, and we apologize for the confusion. The point $\boldsymbol{z}$ refers to the transformed point on the manifold, and it should replace $\boldsymbol{\mu}$ in this expression. Regarding the stability regularization (line 282), $\boldsymbol{z}$ is not directly related to $\boldsymbol{z}_i$ in the regularization term, as $\boldsymbol{z}$ represents a general transformed point in the manifold, while $\boldsymbol{z}_i$ refers to the individual transformed points associated with specific data. We have corrected this inconsistency in the paper.

---

> > > ### Author Response · Authors · 2024-11-24
> > > **[3/3]**
> > >
> > > Question 4: Why not present equations in Section 3 in a more specific form, with the latent manifold $\mathcal{M}$ as $\mathbb{R}^{d}$ with the Euclidean metric?
> > > We believe we addressed the this question in our response to Weakness 2 and kindly refer the reviewer to that response for details. If this explanation is insufficient, please let us know, and we would be happy to try to provide a more satisfying answer.
> > >
> > > Question 5: Why did you choose $d'=1$ for most experiments? Can you add an experiment for $d'\neq1$? How to choose $d'$ in more general cases?
> > > We thank the author for this thoughtful question. For the ARCH, AK and I-FABP datasets we selected $d'=1$ because the intrinsic dimensionality of the data is $d'=1$ which we checked through running isomap. The reviewer is correct in the hypothesis that the datasets are essentially concentrated on a one-dimensional manifold. We included an extra experiment for the rotated swiss roll dataset which is a 2D manifold embedded in 3D. Here we selected $d'=2$, to find the underlying manifold of the swiss roll and see that our method is also effective when $d'\neq 1$.
> > >
> > > Question 6: Computational cost and scalability of the method for large datasets
> > > The reviewer raised a valid concern regarding the computational cost of training isometries, especially when the dataset is large and the method requires iterating over all point pairs, which can be computationally intensive.
> > >
> > > While it is true that computing geodesics for large datasets is challenging, this issue is mitigated to some extent by the ability to precompute distance measures or compute them on the fly using parallel processing across multiple CPUs. However, for very large datasets, the computational complexity can still be prohibitive. This limitation primarily affects data types that require complex distance measures, such as images, where computing geodesics might become a bottleneck.
> > >
> > > That said, our paper focuses on applications with relatively smaller datasets, typical in scientific domains, where interpretability of the latent space is more important than handling vast amounts of data. Therefore, we emphasize that scalability to very large datasets is not the primary goal of our approach, and we acknowledge that further optimizations may be necessary for larger-scale applications.
> > >
> > > Comments on Typos in the Paper
> > > We would like to thank the author reading our paper thoroughly and notifying us about each of these typos. We have adapted the manuscript to remove each one of them.
> > >
> > > We hope these responses address the reviewer’s concerns and improve the clarity and contribution of the manuscript. Thank you again for your valuable feedback.
> > >
> > > Kind regards,
> > > The authors

---

> > > > ### Comment · Reviewer_7H7T · 2024-11-26
> > > >
> > > > Thank the authors for the detailed responses. Based on the replies and the updates in the revised manuscript, I will keep my rating unchanged.

---

> > > > > ### Author Response · Authors · 2024-11-26
> > > > > **Follow-Up on Reviewer Feedback and Invitation for Further Discussion**
> > > > >
> > > > > Dear reviewer,
> > > > >
> > > > > Thank you for your thoughtful review and for taking the time to evaluate and provide feedback on our manuscript. We sincerely appreciate the effort you have put into assessing our work and summarizing its strengths, contributions, and areas for improvement.
> > > > >
> > > > > We have carefully addressed the points you raised and revised the manuscript accordingly. Below is a brief summary of the key updates we made in response to your valuable comments:
> > > > >
> > > > > 1. Generative Modeling and Supporting Claims: We expanded our experiments to provide more robust evidence for the claims made in the abstract and introduction, particularly regarding the generative modeling performance on low-dimensional manifolds. For example, we added new results on the rotated Swiss roll dataset, which highlight the method’s ability to handle more complex, higher-dimensional manifolds. We have also clarified the generative modeling objectives and how they align with isometric learning.
> > > > >
> > > > > 2. Clarifying General vs. Specific Framework: To address concerns about the general formulation in Sections 2 and 3 versus the practical implementation, we have clarified that the experiments are conducted with a Euclidean latent space $\mathbb{R}^d$. We revised the text to emphasize that while the theoretical framework is general, the specific choice of Euclidean space allows us to demonstrate the approach's effectiveness in a common and interpretable setting.
> > > > >
> > > > > 3. Dimension Selection in GRAMPA Dataset: We provided a detailed explanation of how the dimensions $d$ and $d'$ were selected for the GRAMPA dataset, including the use of Isomap to estimate the intrinsic dimensionality and validate our choice. We have included a discussion on the selection of $d$ and $d'$ in the appendix of the manuscript.
> > > > >
> > > > > 4. Computational Scalability: We addressed concerns about the scalability of isometric learning for larger datasets. While our approach is tailored for smaller scientific datasets where interpretability is critical, we included a discussion of computational cost and potential strategies for scaling, such as parallelizing distance computations and leveraging precomputed measures.
> > > > >
> > > > > 5. Typos and Presentation Issues: We corrected all typographical errors and inconsistencies noted in your review and revised the manuscript to improve clarity and flow.
> > > > >
> > > > > We are committed to ensuring the highest quality of our work and would like to ensure we have addressed all of your concerns thoroughly. If there are any remaining questions, doubts, or suggestions for further improvements, we would be grateful to hear them.
> > > > >
> > > > > Please do not hesitate to let us know if there is anything specific you feel requires further clarification or additional evidence. We are open to discussing any aspect of the work and remain dedicated to improving its rigor and impact.
> > > > >
> > > > > Thank you again for your time and constructive feedback, which has been instrumental in enhancing our manuscript.
> > > > >
> > > > > Kind regards,
> > > > > The authors

---

### Official Review · Reviewer_KyCA · 2024-11-04

**Soundness:** 3
**Presentation:** 2
**Contribution:** 2
**Rating:** 3
**Confidence:** 4

**Summary:**

The authors propose a Riemannian flow-matching scheme by learning a metric that pulls the associated interpolations near the support of the given data. The Riemannian metric is induced in the data space by training a diffeomorphism that isometrically maps the data to a latent manifold, preserving the distances of the original data. They propose an objective function that aims to satisfy the properties of the diffeomorphism. In the experiments, they consider a synthetic example to demonstrate the properties of the proposed model and a real-world problem on protein data to highlight its effectiveness.

**Strengths:**

- The technical aspects of the paper seem to be solid.
- The approach for learning a Riemannian metric in the data space is interesting.
- The experiments demonstrate the effectiveness of the proposed model.

**Weaknesses:**

- The paper seems to rely heavily on Diepeveen (2024), with the primary difference being an extension of the diffeomorphism, while the Riemannian flow matching part is a direct application of Chen & Lipman (2024). Although the experiment with protein data is interesting, the technical contribution feels somewhat limited.
- While the writing is technically accurate, the clarity could be improved. For example, adding figures would aid in explaining the modeling approach.
- The objective function is reasonable, but it involves many hyperparameters, which are not straightforward to select (see questions).

**Questions:**

Q1. If I understood correctly, the goal of the diffeomorphism (in the simplest scenario) is to map the data into a Euclidean space of the same dimensionality, where the straight-line distance between data points corresponds to the geodesic distance in the original space (as measured by the Isomap approach). I believe Fig. 1 and Fig. 2 could be improved to aid understanding, or perhaps, adding a new figure.

Q2. The objective function is rather complex and depends on many hyperparameters. A critical quantity, the geodesics between the original data are computed using a graph-based method. While this should work well for the ARCH dataset, we know that graph-based interpolations are less ideal in high-dimensional spaces. Additionally, how do the parameters $\alpha_i$ affect the learned model? I believe an ablation study would have been useful here.

Q3. The data are mapped near a lower-dimensional submanifold, $\mathcal{M}_{d'}$, where dimensionality $d'$ is an additional hyperparameter. How does the choice of $d'$ influence the results? Also, is this achieved by pushing the $d - d'$ dimensions near zero using the "submanifold loss"? I think the diagonal terms in this part of the objective function should be inverted.

Q4. Minor: I think that definitions for $\psi$ (line 229) and the concept of local isometry should be included.

---

> ### Author Response · Authors · 2024-11-24
> **[1/3]**
>
> Dear reviewer,
>
> Thank you for taking the time to read and review our paper. We appreciate your thoughtful feedback, which has helped us identify areas where we can improve the comparison to related work, clarity in our figures and explanations, and the presentation of hyperparameter choices and their impact on the model. Below, we provide detailed responses to your points of feedback and questions.
>
> Strengths
> We appreciate the reviewer’s positive feedback on the technical aspects of the paper and the novelty of our Riemannian metric learning approach, as well as the recognition of the effectiveness of our method in the experiments.
>
> Weakness 1: Comparison to Diepeveen (2024) and Chen & Lipman (2024)
> We agree with the reviewer that that a more thorough explanation of the difference between Diepeveen (2024) and this work should be highlighted in the main body of the text.
>
> In a nutshell, the work by Diepeveen (2024) (i) provides the theory needed for using pullback geometry in a data processing setting, (ii) serves as an initial proof of concept (the work only considers 2D toy data) for learning pullback geometry from a data set and (iii) suggests potential benefits in downstream applications. The paper is fundamentally theoretical in nature, with limited practical applications.
>
> Importantly, the work has two significant methodological limitations that prevent it from realizing its practical potential, making it impossible to demonstrate meaningful downstream applications. These limitations are fundamental: their approach to isometry regularization fails to scale to high-dimensional data, and their chosen architecture lacks the necessary expressiveness. In essence, Diepeveen (2024)'s primary contribution lies in theoretical foundations (i), rather than practical implementation (ii) or applications (iii).
>
> Our work directly addresses these limitations and extends beyond. While building upon their theoretical framework (i), we completely resolve the methodological problem of expressivity by introducing modern diffeomorphism parametrization (through neural ODEs) techniques, in contrast to the Invertible ResNet architecture of Diepeveen (2024). Second, we improve the objective function to achieves both scalability to higher dimensional data and efficient isometry regularization, in contrast to the metric tensor dependent objective function of Diepeveen (2024). As such, our work provides definite proof that pullback geometry can be scaled to real-world data, thus significantly advancing beyond point (ii). Furthermore, we establish an entirely new direction by demonstrating that generation can be made more efficient in a pullback setting - an application that Diepeveen (2024) did not explore. This validates and substantially expands upon their preliminary suggestion in (iii).
>
> To emphasize the difference between Diepeveen (2024) and our work, we rewrote the second paragraph of the related work.
>
> In relation to Chen & Lipman (2024): While we build upon the Riemannian Flow Matching framework of Chen & Lipman (2024), our contribution goes significantly beyond their work by solving a fundamental limitation in generation on manifolds. Specifically: Chen & Lipman (2024) can only perform simulation-free training when closed-form geodesics are available. For general data manifolds, they must fall back to approximate solutions like bi-harmonic geodesics, which:
> a) Impose strong and potentially incorrect geometric assumptions about the data manifold
> b) Require expensive pre-computation of distances before training
> c) Force the use of simulation-based training, resulting in longer training times and more complex dynamics
>
> Our approach fundamentally resolves these limitations by:
> a) Learning closed-form geometric mappings that capture the true structure of the data manifold
> b) Enabling fully simulation-free training for any data manifold
> c) Learning an appropriate metric on the data manifold instead of making assumptions about its geometry
>
> Our method therefore represents a significant advance in making Riemannian Flow Matching practical for real-world data manifolds, rather than being limited to cases where closed-form geodesics are already known. We have clarified this difference in our related work and in Section 3 of the paper.

---

> > ### Author Response · Authors · 2024-11-24
> > **[2/3]**
> >
> > Weakness 2: Clarity and Figures
> > We agree that clarity in writing and the use of intuitive figures can improve the presentation of complex concepts. In response to the reviewer’s comments, we have made the following adjustments:
> >
> > 1. We have added a new example with a figure of isometric learning on the Swiss roll manifold, including both the latent and data manifolds to better visualize the proposed method.
> > 2. We have clarified the mathematical formulation of the paper, including simpler equations for an assumed Euclidean manifold to improve readability in Section 2.
> > 3. We have improved Figure and Table captions to be more descriptive to improve understanding of our experiments.
> > 4. We have adapted explanations regarding our evaluation metrics of generative modeling experiments to promote a more intuitive understanding.
> >
> > If you there are any other specific point in the paper that could be improved or further clarified we would be interested to know them to improve our work.
> >
> > Weakness 3: Hyperparameters in the Objective Function
> > We appreciate the reviewer’s recognition of the challenge posed by the hyperparameters in our objective function. To address this, we included an ablation study in the manuscript, which was part of the version reviewed. We would like to point the reviewer to Table 1, were we evaluate the impact of the hyperparameters $\alpha_2$ (graph matching loss) and $\alpha_4$ (stability regularization) on model performance across two different datasets results for a third dataset can be viewed in Appendix E.
> >
> > Additionally, we would like to clarify that the hyperparameters $\alpha_1$ and $\alpha_3$ were ablated in prior work by Diepeveen (2024), and our findings for these parameters were consistent with those reported in that study. As a result, we did not repeat the ablation experiments for $\alpha_1$ and $\alpha_3$ in this work. We have updated Section 4 to highlight this decision.
> >
> > We hope this additional clarification helps and reassures the reviewer regarding the treatment of hyperparameters in our work.
> >
> > Questions
> > Question 1: Could you add an additional figure to aid in understanding the method?
> > The reviewer correctly understood the goal of the diffeomorphism in the simplest scenario and the use of Isomap for geodesic distances. In response, we have improved the caption of Figure 1 and added a new Figure 2 on isometric learning for the Swiss roll dataset to better illustrate the method. Additionally, we have added the Euclidean versions of the geodesic and distance equations in Section 2.
> >
> > Question 2: How do the parameters affect the learned model and why do you choose to work with graph based methods like isomap for high dimensional data?
> >
> > We believe we addressed the first question in our response to Weakness 3 and kindly refer the reviewer to that response for details. Should further clarification be required, we would welcome the opportunity to provide additional details.
> >
> > Regarding graph geodesics in high dimensions, we agree that this can be prohibitive if unregularized for. In our method, we do regularize for this. The subspace regularization term makes sure that all data points are pushed to a low-dimensional manifold, while the loss that compares pairwise distances now only has to make sure that the distances check out on this low-dimensional space.
> >
> > In addition, this method doesn't rely soley on Isomap geodesics, meaning if you would like to use a different, improved metric for high dimensional data you can use this to train your isometries.
> > Our motivation for the isomap geodesics is precisely because it is a quick and dirty method of calculating geodesic distances (in high dimensions). As can be seen in our experiments and figures, we are able to recover the intended geodesics despite the pitfals of Isomap. Demonstrating the robustness of isometric learning to noisy geodesic distances.

---

> > > ### Author Response · Authors · 2024-11-24
> > > **[3/3]**
> > >
> > > Question 3: Submanifold Dimension
> > > The choice of $d'$ is indeed important. As the method is trying to find the mapping that gives the embedding that methods like isomap generate, there is an optimal dimension that comes out of this. Anything lower or higher will give poor results (see Sonthalia et al., How can classical multidimensional scaling go wrong?, 2021). So to answer the other questions: yes, the first $d'$ dimensions should map to the embedding manifold (with the dimension obtained from isomap-like methods) and the rest is ideally zero. The first part is done by the loss with pairwise distances. The second part is done by the subspace loss. Also, the diagonal has to be this way in order for the remaining components to go to zero. For the theoretical manifolds, of the ARCH dataset and Swiss roll dataset we know the dimension of $d'$ by design. However, for the experimental datasets of the protein dynamics trajectories we do not. Therefore, we used isomap with various embedding dimensions to estimate $d'$. We evaluate the different embeddings through calculation of the stress levels. Stress, in this case, refers to the difference between the pairwise distances in the high-dimensional space and those in the lower-dimensional embedding. Specifically, stress measures the discrepancy between the original pairwise distances $d_{ij}$ in the high-dimensional space and the corresponding distances $\hat{d}_{ij}$ in the embedded lower-dimensional space. Based on this estimate of $d'$ we performed a small hyper parameter search to see which setting of $d'$ gave us the best performance. We included in the Appendix C of the manuscript a discussion on how to select $d'$.
> > >
> > > Question 4: Definitions of $\psi$ and Local Isometry
> > > We are grateful for the reviewer’s suggestion to clarify the definition of $\psi$ and the concept of local isometry. In Section 4, we define $\psi$ as a chart, and we provide an explanation of this differential geometric concept in Section 2 of the manuscript. We also appreciate the reviewer pointing out the absence of a clear definition for local isometry, and we have revised Section 4 to provide a more precise definition.
> > >
> > > We hope these responses address the reviewer’s concerns and improve the clarity and contribution of the manuscript. Thank you again for your valuable feedback.
> > >
> > > Kind regards,
> > > The authors

---

### Official Review · Reviewer_g9xX · 2024-11-06

**Soundness:** 3
**Presentation:** 3
**Contribution:** 2
**Rating:** 5
**Confidence:** 3

**Summary:**

The manuscript is on learning representations tailored for downstream tasks such as interpolation on a lower dimensional manifold and generation. The proposed method uses a Neural ODE to learn a diffeomorphism, which enable simulation free training of dynamics with flow matching.

**Strengths:**

The introductions and background read well, especially I find the method to be well motivated. Overall, the manuscript is clear and the presentation is easy to follow. The authors validate their method on toy and real data, and they conduct an ablation study on various loss terms.

**Weaknesses:**

- The contributions of the paper appear somewhat limited, as it builds upon Diepeveen (2024). I think it would be beneficial if in the main body of the text the authors explained in more details how their work is different from Diepeveen (2024).
- To improve on the current experiments, It would be great to add another toy dataset with known geodesic, e.g. a swiss roll rotated in higher dimensions..
- On line 200, the authors mention that the method requires fewer epochs, but I don't see an experiment backing this claim. Specifically, it could be interesting to compare total training time including learning isometries and dynamics.
- My main concern with the experiments is that there are no comparison with other methods using geometric regularization. The authors mentioned a few in the related work section, but they are not used for comparisons.

**Questions:**

Questions and minor comments:
- I find the notation for the metric $(\cdot,\cdot)^M$ a bit unconventional, maybe $g$ is better? Further, the dependence with $p$ is not clear with the current notation, whereas you could use $g_p$.
- The related work section could be improved. For instance [1,2,3,4] are work that explore geometric regularization of latent space, specifically [1] regularize for geodesics.
- The link with scaling laws in the introduction is a bit confusing. What I understand is that having prior knowledge on the geometry of the data is a great inductive bias, so learning the geometry should be useful.
- On line 200, the authors mention that the method requires fewer epochs and parameters, could you add a reference to the section backing this claim ? Further, does this claim take into account the training of the isometries ?
- On line 287 "we demonstrate the effectiveness of the new regularization terms", what are these new regularization terms ? Is it the graph matching loss and the stability regularization ?
- In section 5.1, for the isometry computation, my understanding is that you need a ground truth geodesic distance. How do you compute it for I-FABP?
- In section 5.3 and 5.4, how many seeds did you use for the experiments ? Could the authors include standard deviation?
- Missing reference on line 1044.


[1] Huguet, Guillaume, et al. "Manifold interpolating optimal-transport flows for trajectory inference." Advances in neural information processing systems 35 (2022): 29705-29718.

[2] LEE Yonghyeon, Sangwoong Yoon, Minjun Son, and Frank C Park. Regularized autoencoders for isometric representation learning. In International Conference on Learning Representations, 2021.

[3] Philipp Nazari, Sebastian Damrich, and Fred A Hamprecht. Geometric autoencoders–what you see is what you decode. 2023.

[4] Fasina, O., Huguet, G., Tong, A., Zhang, Y., Wolf, G., Nickel, M., ... & Krishnaswamy, S. (2023, July). Neural FIM for learning Fisher information metrics from point cloud data. In International Conference on Machine Learning (pp. 9814-9826). PMLR.

---

> ### Author Response · Authors · 2024-11-24
> **[1/3]**
>
> Dear reviewer,
>
> Thank you for taking the time to read and review our paper. We appreciate your thoughtful feedback, which has helped us identify areas where we can improve the papers contributions, experiments, and clarity. Below, we provide detailed responses to your points of feedback and questions.
>
> Strengths
> We are pleased that the reviewer recognized the motivation, clarity, and experimental validation as strengths of our work. Thank you for highlighting these aspects.
>
> Weaknesses 1: Contributions appear limited compared to Diepeveen (2024).
>
> We agree with the reviewer that that a more thorough explanation of the difference between Diepeveen (2024) and this work should be highlighted in the main body of the text.
>
> In a nutshell, the work by Diepeveen (2024) (i) provides the theory needed for using pullback geometry in a data processing setting, (ii) serves as an initial proof of concept (the work only considers 2D toy data) for learning pullback geometry from a data set and (iii) suggests potential benefits in downstream applications. The paper is fundamentally theoretical in nature, with limited practical applications.
>
> Importantly, the work has two significant methodological limitations that prevent it from realizing its practical potential, making it impossible to demonstrate meaningful downstream applications. These limitations are fundamental: their approach to isometry regularization fails to scale to high-dimensional data, and their chosen architecture lacks the necessary expressiveness. In essence, Diepeveen (2024)'s primary contribution lies in theoretical foundations (i), rather than practical implementation (ii) or applications (iii).
>
> Our work directly addresses these limitations and extends beyond. While building upon their theoretical framework (i), we completely resolve the methodological problem of expressivity by introducing modern diffeomorphism parametrization (through neural ODEs) techniques, in contrast to the Invertible ResNet architecture of Diepeveen (2024). Second, we improve the objective function to achieves both scalability to higher dimensional data and efficient isometry regularization, in contrast to the metric tensor dependent objective function of Diepeveen (2024). As such, our work provides definite proof that pullback geometry can be scaled to real-world data, thus significantly advancing beyond point (ii). Furthermore, we establish an entirely new direction by demonstrating that generation can be made more efficient in a pullback setting - an application that Diepeveen (2024) did not explore. This validates and substantially expands upon their preliminary suggestion in (iii).
>
> To clarify these distinctions, we revised the second paragraph of the related work section to highlight these differences more effectively.
>
> Weaknesses 2: Add a toy dataset with known geodesic (e.g., rotated Swiss roll in higher dimensions).
> We agree this is a valuable suggestion and have added experiments using a Swiss roll manifold embedded in 3D. We added visualizations of the data manifold, the learned latent manifold, and geodesic interpolation to illustrate successful isometric learning on a complex nonlinear manifold. Additionally, we included the Swiss roll dataset in our interpolation experiment for which we show the results in Table 2 of the manuscript.
>
> Weaknesses 3:. Backing for the claim on line 200 that the method requires fewer epochs.
> Thank you for pointing this out. We highly value scientific integrity and ensuring that all claims in the paper are fully supported. While we observed during our experiments that our method achieved similar results with fewer epochs due to simplified training dynamics on the manifold, a thorough investigation of this aspect is beyond the scope of this work. To ensure the accuracy of our manuscript, we have decided to remove the statement on line 200 regarding fewer epochs. However, we retain the claim of fewer parameters since it is supported by the results in Table 3. We have added this reference to Section 3 of the paper.

---

> > ### Author Response · Authors · 2024-11-24
> > **[2/3]**
> >
> > Weakness 4: Comparisons with Geometric Regularization Methods
> >
> > We thank the reviewer for highlighting the importance of including comparisons with geometric regularization methods. We agree that such comparisons significantly strengthen the experimental section, and we have carefully addressed this point in our revised submission.
> >
> > To this end, we included results for Geometry Regularized Autoencoders (GRAE), as proposed by Duque et al. (2022). Specifically, we applied GRAE with Isomap and PHATE embeddings as per their methodology, and added the corresponding results in Table 2. This comparison includes all datasets considered in our work, including the newly added Swiss roll experiment. These results demonstrate that while GRAE provides a compelling baseline, our proposed method achieves superior performance across various metrics and datasets, particularly in high-dimensional settings.
> >
> > Additionally, we previously (before submission of the paper) attempted to reproduce two other geometric regularization methods but encountered practical challenges. We would like to share them here to help the reviewer understand our efforts as well as share with the community the challenges encountered in reproducing geometrically regularized methods for high dimensional datasets.
> >
> > 1. Riemannian Auto-Encoders (RAE) (Diepeveen, 2024): The computational complexity of RAE, which arises from the reliance on the metric tensor in the objective function, made it infeasible to reproduce for high-dimensional data.
> > 2. Latent Space Geodesic Learning (Arvanitidis et al., 2017; Arvanitidis et al., 2020): While we made significant efforts to reproduce this approach using the authors’ released code, the method struggled with instability and failed to converge for geodesics in high-dimensional latent spaces. This instability has also been noted in prior works as a limitation of this technique.
> >
> > We believe that our inclusion of GRAE results effectively addresses the reviewer's concerns by providing a meaningful comparison with a state-of-the-art geometric regularization method. These additions further highlight the robustness and scalability of our proposed approach.
> >
> > Thank you again for this valuable feedback, which has enabled us to further refine our experiments and strengthen the contribution of the paper.
> >
> > References:
> >
> > 1. Duque, A. F., Morin, S., Wolf, G., & Moon, K. R. (2022). Geometry regularized autoencoders. IEEE Transactions on Pattern Analysis and Machine Intelligence, 45(6), 7381-7394.
> > 2. Diepeveen, T. (2024). Riemannian Auto-Encoders. Proceedings of the IEEE Conference on Computer Vision and Pattern Recognition (CVPR).
> > 3. Arvanitidis, G., Hansen, L. K., & Hauberg, S. (2017). Latent space oddity: On the curvature of deep generative models. arXiv preprint arXiv:1710.11379.
> > 4. Arvanitidis, G., Hauberg, S., & Schölkopf, B. (2020). Geometrically enriched latent spaces. arXiv preprint arXiv:2008.00565.
> >
> > Questions
> > Question 1: Notation for metric $(\cdot, \cdot)^\mathcal{M}$.
> > We appreciate the reviewer’s suggestion to align our notation more closely with standard machine learning practices. However, we use $(\cdot, \cdot)^\mathcal{M}$ as it is consistent with coordinate-free Riemannian geometry and aligns with the notation in Diepeveen (2024).
> >
> > To address the reviewer’s concern about clarity, we have added subscripts to explicitly denote the point $\mathbf{p}$ where the inner product is taken, e.g., $(\cdot, \cdot)_{\mathbf{p}}^\mathcal{M}$. This adjustment improves consistency and clarity while preserving the benefits of the coordinate-free notation.
> >
> > Question 2: Related work section improvements.
> > Thank you for pointing out additional relevant work on geometric regularization of latent spaces. We have reviewed the papers you mentioned and updated the related work section accordingly. Specifically, we have incorporated references to the works on geometric regularization, including the geodesic regularization method in (Huguet et al., 2022), and will ensure that the relevant references are included in the camera-ready version. Additionally, we have added a geometrically regularized latent space method (Duque et al.,2022) to our interpolation experiments and included this work in the updated related work section.
> >
> > Question 3: Clarify the link to scaling laws.
> > The reviewer’s understanding is correct: having prior knowledge of the geometry of the data acts as a strong inductive bias, leading to better scalability. We have revised the introduction to use the term "inductive bias" explicitly to make this connection clearer.
> >
> > Question 4: Could you provide a reference for the statement that the method requires fewer epochs and parameters?
> > We believe we addressed this question in our response to Weakness 3 and kindly refer the reviewer to that response for details. If this explanation is insufficient, please let us know, and we would be happy to try to provide a more satisfying answer.

---

> > > ### Author Response · Authors · 2024-11-24
> > > **[3/3]**
> > >
> > > Question 5: Specify "new" regularization terms.
> > > The "new" regularization terms refer to the graph matching loss and stability regularization introduced in this paper. We have removed the ambiguous term "new" and explicitly named these contributions to avoid confusion.
> > >
> > > Question 6: Geodesic distances for I-FABP.
> > > We assume a locally Euclidean geodesic distance on the data manifold $\mathcal{D}$, approximated using Isomap (Tenenbaum et al., 2000), for the I-FABP protein trajectory and other datasets. This was included in the caption of the ablation study table (Table 1). We improved the wording of the caption to clarify the text further.
> > >
> > > Question 7: Number of seeds and standard deviations.
> > > We reported results for a single random seed in the initial submission and the standard deviations provided in Table 2 are over the 100 evaluated geodesics. As authors we care about reproducibility of our results and thank the reviewer for pointing out how we could improve this through inclusion of standard deviations of models across multiple seeds. We have adjusted Table 1 and 2 of the results to include standard deviations over three selected seeds. The standard deviations included in Table 2 are across 300 geodesics (100 geodesics for each seed). For the generation experiments we will continue our efforts and hopefully we can include standard deviations before the rebuttal period ends, if not we will include them in the camera ready version of the paper.
> > >
> > > Minor Comment: Missing Reference
> > > Thank you for catching the missing reference on line 1044. We corrected this in the revised manuscript.
> > >
> > > Once again, we thank the reviewer for their constructive feedback and valuable suggestions. We look forward to hearing your thoughts on these updates.
> > >
> > > Kind regards,
> > > The authors

---

> ### Author Response · Authors · 2024-11-26
> **Manuscript Update to Include Swiss Roll and Multiple Seed Generation Experiments**
>
> Dear Reviewer,
>
> Thank you for your feedback and suggestions regarding our experiments. In response to your feedback on running experiments with multiple seeds and your suggestion to include results for the Swiss roll manifold rotated in higher dimensions, we have further updated the manuscript as follows:
>
> 1. Multiple Seed Experiments: To address your feedback, we have rerun the generative modeling experiments in Table 3 using 3 seeds. Results are now averaged, and standard deviations are reported to ensure robustness and reproducibility.
> 2. Swiss Roll Experiments in Higher Dimensions: In line with your suggestion, we have added new experiments on the Swiss roll manifold rotated in higher dimensions. We previously only included results through Figure 2 and Table 2, but now we have also included results on this manifold in Table 3.
> These updates also highlight our method’s ability to handle data manifolds with complex, non-uniform curvature, further validating its performance.
>
> We sincerely appreciate your valuable input, which has helped enhance the quality and depth of our work.
>
> Kind regards,
> The authors

---

### Official Review · Reviewer_RFZZ · 2024-11-20

**Soundness:** 2
**Presentation:** 3
**Contribution:** 2
**Rating:** 3
**Confidence:** 4

**Summary:**

The paper introduces Pullback Flow Matching (PFM), a novel generative modeling framework designed to operate on data manifolds, which uses pullback geometry and isometric learning to preserve the geometric structure of data manifolds while enabling efficient generation and precise interpolation in latent spaces. PFM leverages Neural Ordinary Differential Equations (Neural ODEs) to improve the expressiveness and efficiency of learning diffeomorphisms. This approach enables more accurate interpolation and generation of data. The paper demonstrates the effectiveness of PFM through experiments on synthetic datasets, molecular dynamics simulations, and protein sequence data. The results show that PFM can generate novel proteins with desired properties, indicating its potential for real-world applications in scientific domains where precise modeling of complex systems is required.

**Strengths:**

- This paper demonstrates strong mathematical foundations and theoretical rigor, which helps ensure the method’s reliability and provides theoretical guarantees for its performance.
- The neural ODE-based diffeomorphisms and design of product manifold eliminates need for expensive Riemannian metric tensor calculations and requires only geodesic distance approximations.

**Weaknesses:**

- While PFM is presented as an improvement over RFM, there’s no quantitative evidence showing the relative performance. The paper does not provide direct experimental comparisons between PFM and RFM.
- The paper mainly compare PFM with CFM and VAE, which misses comparasion with more state-of-the-art models, such as Diffusion, GAN, SLERP.
- While the paper claims geometric preservation through pullback geometry and isometric learning, no formal theoretical proof is provided showing that PFM actually preserves geometric structures. The experiments are insufficient to verify geometric preservation claims as well.

**Questions:**

- The notations can be more clear. For example, the scheduler $κ(t)$ and vector field $v(t)$ are not clearly defined in the main body of the paper. This may lead to misunderstandings of the content of the article.
- The article will be more convincing if experiments that compare PFM with RFM are included.
- How might the properties of different manifolds, particularly those with non-trivial curvature, affect the performance of PFM?

---

> ### Author Response · Authors · 2024-11-26
> **[1/3]**
>
> Dear reviewer,
>
> We sincerely thank you for your time and thoughtful feedback on our work. We appreciate your recognition of the strong mathematical foundations and theoretical rigor of our method. Below, we address the specific weaknesses, questions, and suggestions you raised.
>
> Weaknesses
> Weakness 1: Lack of quantitative comparisons with RFM
> We appreciate the reviewer’s suggestion regarding quantitative comparisons with RFM and acknowledge the importance of clearly differentiating our approach from that of Chen & Lipman (2024). Below, we explain how our contributions address the limitations of RFM and why we chose not to include direct experimental comparisons in this work.
>
> While we build upon the Riemannian Flow Matching (RFM) framework proposed by Chen & Lipman (2024), our work addresses significant challenges that limit the practicality of RFM for general data manifolds. Chen & Lipman (2024) propose two approaches for handling data manifolds:
>
> Assuming a predefined geometry for the data manifold
> This approach relies on strong assumptions about the manifold’s geometry, which are often unrealistic for real-world datasets where the true geometry is unknown or highly complex. These assumptions can lead to inaccuracies in geodesic calculations and modeling.
> Using biharmonic geodesics as an approximation
> This approach avoids direct assumptions but requires expensive pre-computation of distances and simulation-based training, which are computationally intensive and limit scalability.
> These limitations impact RFM in two key ways:
>
> Assumption of Geometry: Predefined geometries may fail to capture the true structure of real-world data, potentially biasing the learned representations.
> Simulation-Free Training: The reliance on simulation-based methods increases complexity and training time, which are critical challenges for generative modeling tasks.
> Our contributions address these limitations by:
>
> 1. Learning a closed-form mapping that preserves the intrinsic structure of the data manifold, eliminating the need for predefined geometries or approximations.
> 2. Enabling simulation-free training through the pullback metric, ensuring efficient and scalable learning even for high-dimensional and complex manifolds.
> 3. Learning an appropriate metric rather than relying on predefined or approximate geodesics, providing robustness and flexibility across diverse datasets.
> These advances make RFM more practical for general data manifolds and enable principled generative modeling without requiring restrictive assumptions or complex training dynamics.
>
> Direct experimental comparisons with RFM were not performed because its reliance on predefined geometries or biharmonic geodesics is fundamentally misaligned with our goals of addressing general data manifolds without restrictive assumptions. Our method is specifically designed to address the challenges of general data manifolds and to provide simulation-free training, which are not achievable with RFM as formulated.
>
> To address this concern, we revised Section 3 of the paper to better clarify the distinctions between RFM and our approach, and we elaborated on the limitations of RFM in simulation-free training and general manifold modeling. These updates aim to clarify our contributions and highlight why a direct experimental comparison was not included.
>
> We sincerely thank the reviewer for their thoughtful feedback, which helped us improve the manuscript and presentation of our work.
>
> Weakness 2: Comparison with other state-of-the-art models (e.g., Diffusion, GAN, SLERP)
> We thank the reviewer for highlighting the importance of baselines in contextualizing performance. We recognize the importance of comparisons with other generative models. However, this work is dedicated to advancing flow-matching methods, and our experiments were specifically designed to deepen insights within this framework.
>
> Our approach is generalizable and could be applied to other generative models, such as diffusion models, by leveraging our learned latent manifold. However, including such experiments would shift the focus to a broader comparison between flow-matching techniques and other generative modeling paradigms, which is beyond the scope of this paper. Instead, we prioritized providing insights and improvements within the family of flow-matching methods, as these are most relevant to our contributions.
>
> In response to feedback from reviewers, we incorporated additional baselines in Table 2, comparing our approach with geometric regularization methods such as GRAE. These experiments align with our focus on isometric learning and regularization while addressing the broader need for meaningful baselines.
>
> We hope this explanation clarifies our decisions and reasoning about experimental design and inclusion. We value this discussion and appreciate the opportunity to share our thought process on this matter.

---

> ### Author Response · Authors · 2024-11-26
> **[2/3]**
>
> Weakness 3: Claims of Geometric Preservation
> Thank you for raising this important concern. We value scientific integrity and aim to ensure that all claims are well-supported by both theoretical and empirical evidence.
>
> The geometric preservation claims are grounded in the pullback metric framework established by Diepeveen (2024), which ensures the approximation of geodesics between the data manifold ($\mathcal{D}$) and the latent manifold ($\mathcal{M}$). Building on this, our method supervises the preservation of pairwise distances to maintain manifold structure.
>
> We validate these claims through the following experimental results:
> 1. Figure 2: Demonstrates isometric learning for the rotated Swiss roll, where geodesics on the data manifold correspond to shortest paths on the learned latent manifold.
> 2. Figure 3: Shows generative trajectories for the ARCH dataset, highlighting how our learned latent submanifold aligns with the data manifold geometry during generative tasks.
> 3. Figure 5: Evaluates geodesic interpolation on the ARCH dataset, comparing our method to Isomap geodesics and demonstrating alignment with the true submanifold.
> 4. Table 1: Provides an ablation study of regularization terms, showing that isometry and low-dimensionality metrics improve when both Graph Matching and Stability regularizations are included.
> 5. Table 2: Quantifies geodesic preservation via RMSE on the 100 longest Isomap geodesics across multiple datasets, showing that our method outperforms GRAE and ($\beta$)-VAEs.
> 6. Table 3: Highlights downstream generative performance, where our approach achieves better latent space dimensionality and $1$-NN accuracy, reflecting improved generative quality.
>
> To address concerns about geometric preservation, we revised Figures 2 and 3 to better demonstrate isometric learning and generative trajectories. Additionally, we updated Table 2 and added detailed results on the Swiss roll dataset to showcase performance on curved manifolds with non-trivial geometry. These updates provide empirical evidence supporting our geometric preservation claims.
>
> We would appreciate the reviewer’s input on whether our current approach sufficiently addresses their concerns or if there are specific aspects they feel could be better demonstrated. We welcome any suggestions for improvements or directions for future work that could further strengthen our claims or broaden the scope of this research.
>
>
> Questions
> Question 1: Clarity of notation ($\kappa(t)$, $v(t)$)
> hank you for pointing out the need for clearer notations. In response, we have made the following improvements to enhance clarity and ensure the equations are more intuitive:
> 1. We have added a description of the vector field $v_t$ in Section 3.
> 2. We have simplified the notation of Section 3 by excluding $\kappa (t)$ to make the equations more concise and accessible.
> For readers interested in more detailed equations and descriptions, including the role of $\kappa(t)$, we have provided these in Appendix A. We hope these changes make the presentation clearer and easier to follow.
>
> Question 2: Inclusion of comparison to RFM
> We have addressed this concern in our response to Weakness 1 and have made efforts to clarify our approach there. If this response does not fully resolve your concerns, we would greatly appreciate feedback on any remaining questions or suggestions for how we could further improve the manuscript to better address this point.

---

> ### Author Response · Authors · 2024-11-26
> **[3/3]**
>
> Question 3: Impact of manifold properties, especially non-trivial curvature
> We thank the reviewer for raising the insightful question regarding the impact of curvature on the performance of PFM. Manifold curvature can significantly influence data properties, and understanding its impact is important for generative methods operating in curved spaces.
>
> A full investigation of the impact of curvature across complex manifolds on the Pullback Flow Matching framework is outside the scope of this work. However, we agree that this is an important direction for future research. Our work primarily focuses on demonstrating the versatility of PFM across empirically relevant datasets rather than performing a comprehensive theoretical analysis of manifold curvature.
>
> Regarding the protein trajectory datasets, the curvature of these manifolds is not explicitly known.
> This uncertainty means we cannot directly assess how curvature impacts results in this setting. Nonetheless, our method shows strong empirical performance on these datasets, suggesting robustness to the unknown manifold properties.
>
> To provide additional insights, we included experiments on a manifold with known complex non-uniform curvature: the Swiss roll dataset. The newly added results in Table 2 and Table 3 demonstrate the following:
> 1. Learning isometries and interpolation: Table 2 highlights our method’s ability to learn isometries for the curved Swiss roll manifold and to perform accurate interpolations, as illustrated in the newly added Figure 2.
> 2. Accurate generation and simulation-free training: Table 3 shows that PFM can leverage its learned isometry to generate data accurately on this curved manifold and to enable simulation-free training.
>
> We hope these additional results address your concerns and showcase the applicability of our method to manifolds with non-trivial curvature.
>
> Once again, we thank the reviewer for their constructive feedback, valuable suggestions and their contribution to improving the manuscript. We look forward to hearing your thoughts on these updates.
>
> Sincerely,
>
> The authors

---

### Author Response · Authors · 2024-12-04
**Summary of Reviewer Feedback and Our Responses [1/2]**

We sincerely thank the reviewers for their thoughtful and detailed feedback, which has played a pivotal role in improving the clarity, rigor, and overall quality of our manuscript. This summary aims to consolidate the key concerns raised during the review process, detail our corresponding responses, and highlight the resulting improvements. We deeply value the reviewers’ insights, which have not only enhanced the presentation and scope of our work but have also provided meaningful directions for its future development. Below, we present an aggregated summary of the primary issues identified and the measures we took to address them.

1. Differentiation from Diepeveen (2024) and Chen & Lipman (2024):
   - Reviewer Concern: Our contributions were not clearly differentiated from these prior works.
   - Our Response: We clarified that while Diepeveen (2024) introduced foundational theory for pullback geometry, it is inherently limited in practical applicability. Specifically, their approach fails to scale to high-dimensional data due to their reliance on invertible ResNets and a metric tensor-dependent objective function. In contrast, we introduced a modern diffeomorphism parameterization via Neural ODEs and a redesigned objective that supports scalable and efficient isometry regularization. This enables our method to achieve robust performance on high-dimensional and real-world datasets, which was not feasible in Diepeveen’s framework. Moreover, our work explores entirely new applications, such as simulation-free generative modeling, which Diepeveen only speculated as a potential future direction.
   Similarly, while we build on the Riemannian Flow Matching (RFM) framework introduced by Chen & Lipman (2024), we overcome a fundamental limitation in their method: the reliance on predefined geometric assumptions or precomputed geodesics. Chen & Lipman (2024) can only perform simulation-free training when closed-form geodesics are available. For general data manifolds, their reliance on approximate solutions (e.g., bi-harmonic geodesics) introduces inaccuracies, adds computational overhead, and negates the benefits of simulation-free training. Our approach resolves these issues by learning closed-form geometric mappings directly from data, enabling true simulation-free training on any data manifold without restrictive geometric assumptions.
    These distinctions, along with supporting experiments in Section 5, are emphasized in the revised related work in Section 1 and improved methods sections in Section 3 and 4. We believe these advancements clearly demonstrate the practical and theoretical contributions of our work beyond prior studies.

2. Geometric Preservation and Comparison with Geometric Regularization Methods
    - Reviewer Concern: The validity of our claims regarding geometric preservation was questioned due to the absence of formal theoretical proof and perceived insufficiency of experimental evidence. Additionally, reviewers noted the lack of comparisons with other geometric regularization methods cited in the related work section.
    - Our Response: Our geometric preservation claims are built on the solid theoretical foundation provided by Diepeveen et al. (2024), which rigorously establishes the pullback metric framework and its ability to ensure geodesic alignment between data and latent manifolds. As this framework has already been theoretically proven, our work did not require additional proofs. Instead, we extended this foundation by supervising the preservation of pairwise distances to maintain manifold structure and demonstrated its practical effectiveness through extensive empirical validation on high dimensional real-world datasets.
    Empirically, we validated these claims using the Swiss roll dataset (Figure 2, Table 2) to demonstrate isometric learning, geodesic interpolation (Figure 5) on the ARCH dataset to showcase alignment with manifold geometry, and ablations (Table 1) to highlight the impact of key regularizations. To address the lack of comparisons, we included results for Geometry Regularized Autoencoders (GRAE) using Isomap and PHATE embeddings (Table 2), showing our method's superior performance, particularly in high-dimensional settings. Challenges in reproducing other methods, including Riemannian Autoencoders (RAE) and Latent Space Geodesic Learning, were transparently documented, highlighting computational and stability issues.

---

> ### Author Response · Authors · 2024-12-04
> **Summary of Reviewer Feedback and Our Responses [2/2]**
>
> 3. Hyperparameter Selection and Ablation Study:
>     - Reviewer Concern: The complexity of the objective function and the challenge of selecting hyperparameters were questioned.
>     - Our Response: We acknowledged the reviewer’s concern and highlighted the ablation study included in our manuscript (Table 1, Appendix E), which evaluates the impact of key hyperparameters (\(\alpha_1\) for graph matching loss and \(\alpha_3\) for stability regularization) across multiple datasets. Results confirm their influence on model performance and provide guidance for selection. Additionally, we clarified that the hyperparameters \(\alpha_2\) and \(\alpha_4\) were thoroughly analyzed in prior work by Diepeveen et al. (2024), and our findings align with their results, eliminating the need for repeated ablations. This approach was emphasized in the revised Section 4 for greater clarity.
>
> 4. Curved Manifold Evaluation via Swiss Roll:
>    - Reviewer Concern: The impact of manifold curvature on performance was highlighted, with requests for evaluation on curved manifolds.
>    - Our Response: We conducted additional experiments on the Swiss roll dataset rotated in higher dimensions, which incorporates non-trivial curvature. Results (Figures 2, Table 2 and Table 3) confirm the robustness of our method, demonstrating effective isometric learning and interpolation on curved manifolds.
>
> 5. Additional Seeds in Experiments:
>    - Reviewer Concern: The robustness of our results needed validation through multiple random seeds.
>    - Our Response: Experiments were re-evaluated using three random seeds, and standard deviations were reported for all key results in Tables 1 and 2. Multi-seed results for generative modeling experiments were also added in Table 3 to confirm robustness.
>
> 6. Dimensionality Selection for Datasets:
>    - Reviewer Concern: Clarity was requested on how the submanifold dimensions ($d'$) were determined.
>    - Our Response: We explained that $d'$ was chosen based on manifold embedding methods (e.g., Isomap) and validated using stress metrics. Details were added to Appendix C, including discussions on hyperparameter search for unknown manifolds.
>
> 7. Method Clarity and Presentation:
>    - Reviewer Concern: The explanation of the method, notation, and figures required improvement.
>    - Our Response: We added clearer definitions (e.g., for $\psi$ and local isometry), improved mathematical notation, and introduced new figures to illustrate isometric learning and results (e.g., Swiss roll visualizations). Figure captions and manuscript text were revised for clarity.
>
> We believe we have effectively addressed all reviewer concerns through detailed responses, additional experiments, and clarifications. These revisions, including new comparisons and enhanced explanations, provide stronger support for our claims and clearly highlight the contributions of our work. We are confident that the manuscript now more accurately reflects the robustness of our approach and its distinctions from prior work. We appreciate the reviewers' valuable feedback, which has significantly contributed to improving the quality of our research. For further details, we refer the reader to the comprehensive responses provided above.
>
> We would like to express our sincere gratitude to the Area Chair for their part in the review process and for their consideration of our work.
>
> Kind regards,
>
> The authors

---

### Meta-Review · Area_Chair_gZG7 · 2024-12-18

**Metareview:**

This paper presents Pullback Flow Matching (PFM), a generative modeling framework for data manifolds that uses pullback geometry and isometric learning to preserve manifold geometry. By enhancing isometric learning with Neural ODEs and a scalable training objective, PFM is argued to improve manifold learning and generative performance. The effectiveness of this method is demonstrated through experiments on synthetic data, molecular dynamics data, and experimental peptide sequences.

The paper received average scores between 3 and 5, with critics concentrating on lacks of novelty with respect to reference Diepeveen (2024) and Chen & Lipman (2024), lack of experimental validation with respect to other manifold based generative approaches and lack of theoretical guarantees about preservation of isometries. In this light, I am recommending a reject decision, and I encourage the authors to further strengthen their work on the questions raised by reviewers.

**Additional Comments On Reviewer Discussion:**

After the rebuttal, only few discussion happened between authors and reviewers, despite authors engaging with constructive feedbacks on the reviews.

---

### Decision · Program_Chairs · 2025-01-22

Reject